# Drivers for optimum sizing of wind turbines for offshore wind farms

Mihir Mehta, Michiel Zaaijer, and Dominic von Terzi

Wind Energy Group, Faculty of Aerospace Engineering, Delft University of Technology, Netherlands

**Correspondence:** Mihir Mehta (m.k.mehta@tudelft.nl)

**Abstract.** Large-scale exploitation of offshore wind energy is deemed essential to provide its expected share to electricity needs of the future. To achieve the same, turbine and farm-level optimizations play a significant role. Over the past few years, the growth in the size of turbines has massively contributed to the reduction in costs. However, growing turbine sizes come with challenges in rotor design, turbine installation, supply chain, etc. It is, therefore, important to understand how to size wind turbines when minimizing the Levelized Cost of Electricity (LCoE) of an offshore wind farm. Hence, this study looks at how the rated power and rotor diameter of a turbine affect various turbine and farm-level metrics and uses this information in order to identify the key design drivers and how their impact changes with setup. A Multi-disciplinary Design Optimization and Analysis (MDAO) framework is used to perform the analysis. The framework uses low-fidelity models that capture the core dependencies of the outputs on the design variables while also including the trade-offs between various disciplines of the offshore wind farm. The framework is used, not to estimate the LCoE or the optimum turbine size accurately, but to provide insights into various design drivers and trends. A baseline case, for a typical setup in the North Sea, is defined where LCoE is minimized for a given farm power and area constraint with the IEA 15 MW reference turbine as a starting point. It is found that the global optimum design, for this baseline case, is a turbine with a rated power of 16 MW and a rotor diameter of 236 m. This is already close to the state-of-the-art designs observed in the industry and close enough to the starting design to justify the applied scaling. A sensitivity study is also performed that identifies the design drivers and quantifies the impact of model uncertainties, technology/cost developments, varying farm design conditions, and different farm constraints on the optimum turbine design. To give an example, certain scenarios, like a change in the wind regime or the removal of farm power constraint, result in a significant shift in the scale of the optimum design and/or the specific power of the optimum design. Redesigning the turbine for these scenarios is found to result in an LCoE benefit of the order of 1-2% over the already optimized baseline. The work presented shows how a simplified approach can be applied to a complex turbine sizing problem, that can also be extended to metrics beyond LCoE. It also gives insights to designers, project developers, and policy makers as to how their decision may impact the optimum turbine scale.

**Keywords:** Offshore wind, Wind turbine design, Wind farm design, Multi-disciplinary design optimization and analysis (MDAO), Levelized Cost of Electricity (LCoE).

## 1 Introduction

The role of electrification of sectors is essential for tackling climate change where most electricity has to come from low-cost renewables, mainly wind and solar. Wind energy is expected to provide a third of the expected electricity needs by 2050 with an installed capacity of about 6000 GW (International Renewable Energy Agency (IRENA), 2019). The expected share of wind can only be achieved by exploiting the ocean area offshore with relatively steadier and higher wind speeds. This scale also demands continuous innovation and cost-reduction strategies. The Levelized Cost of Electricity (LCoE) of offshore wind for some upcoming farms in the Dutch waters, without the grid connection, is already close to 50 €/MWh (Lensink and Pisca, 2019; Wind & water works, 2022). These cost reductions in offshore wind can be largely attributed to upscaling of turbines, and declining Operations and Maintenance (O&M) costs (Lantz et al., 2012; International Renewable Energy Agency (IRENA), 2019; Veers et al., 2019). Optimization of turbines and wind farms to achieve further cost reductions is crucial in achieving the intended scale of deployment.

Over the years, optimization methods and metrics have evolved and gotten better. Andrew Ning et al. (2014) and Chehouri et al. (2015) discuss how various objective functions and constraints lead to different rotor designs. Initially, the focus was on the aerodynamic performance of the blade in order to maximize the power coefficient ($c_P$) of the rotor. However, this metric would ignore the mass of the rotor involved, which was solved by minimizing the ratio of mass to the Annual Energy Production (AEP). Although promising, this metric would not take into account the costs of various components. The Cost of Energy (CoE) solved this issue as it involved both the costs and the AEP. However, in an offshore wind farm, there are various disciplines interacting with each other, and the CoE of the turbine alone would not be a comprehensive metric anymore. Later on, LCoE became the most widely adopted metric for optimization studies (Dykes, 2020). LCoE is a metric that is easy to calculate, covers all the aspects of a wind farm, and is hence universal in nature. Various wind farms across different sites or even different technologies could be compared simply by looking at the LCoE values.

To achieve further cost reductions, the benefits of systems engineering by using Multi-disciplinary Design Analysis and Optimization (MDAO) have also been explored by Ashuri et al. (2016), Perez-Moreno et al. (2018), Dykes et al. (2018), and Bortolotti et al. (2022). An MDAO-based approach captures the trade-offs between various disciplines of a system and results in a better design, compared to traditional sequential optimization. The studies also point out the importance of using the overall LCoE of the wind farm as the global objective function. Bortolotti et al. (2019) developed reference wind turbines for onshore and offshore applications using such an MDAO-based framework. Dykes et al. (2018) and Serafeim et al. (2022) explored the optimization of the rotor for a turbine with a fixed rated power using the LCoE of the farm as the objective function. Most studies related to turbine optimization in a farm setting keep the rated power fixed and/or rotor diameter fixed, and the effect of upscaling the turbine itself is often not the focus. Ashuri et al. (2016) optimized a 5 MW reference turbine and scaled it up to 10 MW and 20 MW to evaluate the effect on LCoE and find an increasing LCoE trend with upscaling. However, the costs for Balance of System (BoS) and O&M are assumed to scale with the rated power, with a fixed value for the exponent. In reality, the interactions of the turbine with the other elements of the farm are much more complex and require modeling of all the disciplines of the wind farm. Sieros et al. (2012) performed an upscaling study for turbines in the range of 5-20 MW, with

constant specific power, using classical similarity rules. The results showed an increase in the levelized production cost with turbine scale, for the same technology level. However, the focus of the study was on a simplified upscaling method, especially for the turbine, while the models for the rest of the wind farm were expressed simply as a percentage of turbine costs. Shields et al. (2021) studied the impact of turbine upscaling and plant upsizing on various farm-level parameters providing several valuable insights. They find a reduction in LCoE by up to 20% when upscaling turbines from 6 to 20 MW and upsizing the farm from a 500 MW capacity to a 2500 MW capacity. However, the study assumes a fixed cost per kW for the turbines and also limits the specific power of the turbines when upscaling.

The limitations in previous work w.r.t. the turbine design space and turbine costs are expected to have a significant impact on the generalization of the results and conclusions. Both the numerical findings and the insights into drivers for turbine scaling will be affected. The work presented in this paper aims to capture, more comprehensively, the variations in the turbine design and costs when scaling turbines, while also including the interactions and trade-offs occurring at a farm level. The main research question this study tries to answer is as follows:

*What drives the sizing of wind turbines for minimum LCoE of offshore wind farms?*

The question can be further broken down into four sub-questions:

1. For a typical case, how does the turbine scale drive various trade-offs at a farm level, and what is the optimum turbine size?

2. How do uncertainties, technology changes, and economic conditions drive the optimum turbine design?

3. How do various farm design conditions drive the optimum turbine design?

4. How do farm-level constraints drive the optimum turbine design?

The turbine size refers to the two main defining variables of the turbine, rated power and the rotor diameter. The two variables are optimized w.r.t. the LCoE of a hypothetical wind farm, using an MDAO framework that includes low-fidelity models for every discipline of an offshore wind farm. The findings of this work may inform policy-makers and wind farm developers with useful insights. However, the implementation is simplified and the chosen set of design variables is limited. Thus, this study aims to be exploratory work that provides the potential possibilities of application of MDAO in large-scale wind farm design problems.

## 2   Methodology and setup

This section discusses the generic modeling approach along with the problem formulation, models for various farm disciplines, model inputs, and the case study being explored in this research.

## 2.1 Overview and rationale of the approach

The problem of optimizing the turbine size for an offshore wind farm is complex, as changing the key specifications of the turbine impacts all elements in the farm. For instance, a change in the rated power of the turbine changes the current in the infield cables, and hence, cabling costs. If the farm power is given, changing the turbine rated power changes the number of turbines in the farm having a significant impact on O&M costs, installation costs, wake losses, etc. Similarly, any change in the rotor diameter affects the power and thrust curve of the turbine, impacting the support structure design, wake losses, etc. Hence, both key parameters of the turbine significantly affect both costs and AEP of the wind farm. For a turbine sizing problem, capturing the essential trade-offs at a wind farm level is paramount, making the use of an MDAO-based approach that includes all disciplines in the wind farm, inevitable.

The study does not focus on the development of an MDAO-based framework per se but rather uses the framework as an analysis block to evaluate the LCoE of the farm for a given turbine configuration. The framework will be used to perform analyses that provide insights into the fundamentals of optimal turbine sizing. Some studies that applied MDAO to a turbine-optimization problem for a wind farm, along with the missing dependencies, are discussed in Section 1. The requirements of the model-fidelity for each discipline of the wind farm depend on the purpose of the study. For a turbine-sizing study with turbine rated power (P) and rotor diameter (D) as the design variables, it is key that the models for any given discipline respond correctly to the change in the design variables, directly or indirectly. For instance, an increase in the rated power results in a decrease in the number of turbines (if the farm power is kept constant), and as a consequence, results in lower O&M and installation costs. It is essential for the O&M and BoS models to capture these trends reasonably well. However, a model that assumes O&M costs to be a function of the farm rated power or a function of the turbine rated power like in Ashuri et al. (2016), fails to capture the necessary trade-offs. Similarly, the turbine costs (including the support structure) change non-linearly w.r.t. changes in both the rotor diameter and rated power of the turbine. However, a model that scales the turbine costs linearly with the rated power, like in Shields et al. (2021), does not capture the variations in turbine costs because of changes in the rotor diameter. This would significantly impact the findings and conclusions. Hence, it is crucial that the models for all the disciplines in the wind farm capture the dependencies on the design variables. Having low-fidelity models that can capture the essential trade-offs allows the user to quickly evaluate hundreds of turbine designs. The purpose of the MDAO framework, in the context of this research, is not the accurate estimation of LCoE or the optimum design. The main purpose of the framework is to serve as an analysis block that captures the dependencies of various wind farm elements on the design variables and, hence, can be used in identifying the key drivers of turbine sizing. The drivers could be in the form of technology changes, farm conditions, or even policy-level changes, all of which could be identified and quantified with such a comprehensive framework. A summary of the key elements of the approach is given below.

- Model lowest necessary fidelity required for all wind farm disciplines

- Capture direct and indirect dependencies of each discipline on the design variables

- Capture interactions between different wind farm disciplines

– Analyze and visualize the response surface of the outputs

        – Identify key drivers of turbine sizing by analyzing the sensitivity of the outputs to various inputs

In line with these considerations, the next sections first describe the MDAO framework and the optimization problem that is addressed with this. The subsequent descriptions of the models focus on the dependencies that are identified to be relevant for this study, rather than on comprehensive mathematical descriptions.

## 2.2 Modelling and optimization framework

For this research, the MDAO-based framework developed by Tanmay (2018) and Sanchez Perez Moreno (2019) is expanded and updated. This framework is shown in Fig. 1, where all the disciplines of a wind farm are modeled and coupled via coupling variables. The software is open source and can be accessed via the repository of Mehta (2023). The framework uses certain user and modeling inputs, highlighted by the green blocks. For the optimization process, the design variables are assigned values
by the optimizer in each function evaluation, in which the objective function and the constraints are evaluated.

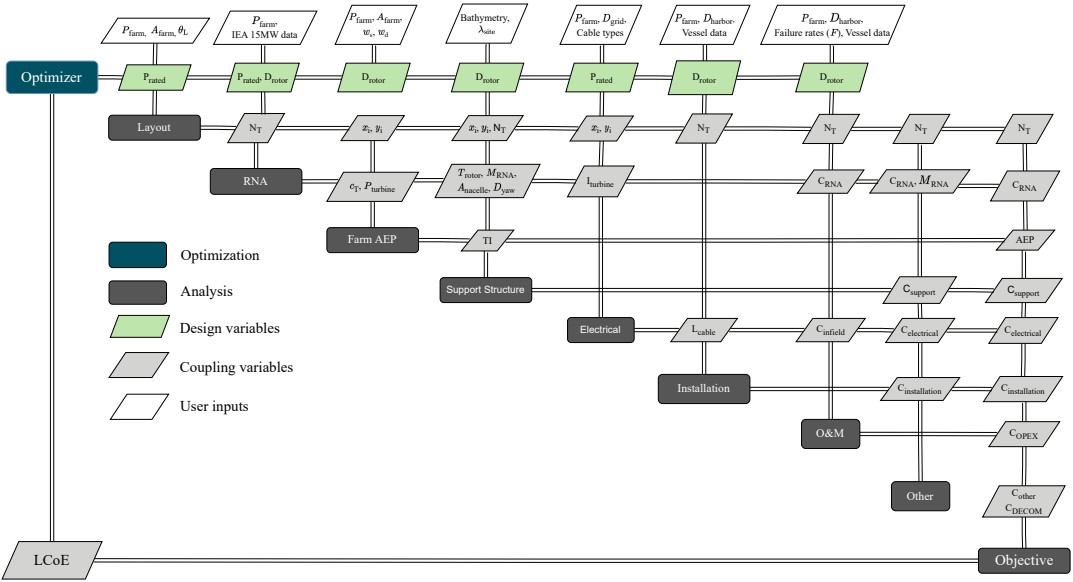

**Figure 1.** eXtended Design Structure Matrix (XDSM) of the MDAO framework.

## 2.3 Problem formulation

This section discusses the formulation of the optimization problem. The baseline problem is formulated as given in Eq. (1) where the objective function is the LCoE of the offshore wind farm which is to be minimized w.r.t the rated power ($P$) and the rotor diameter ($D$) of the turbine.

$$\min_{P,D} \quad \text{LCoE}$$

$$\text{s.t.} \quad P_{\text{farm}} = 1\,\text{GW} \tag{1}$$

$$\text{s.t.} \quad A_{\text{farm}} = 150\,\text{km}^2$$

An equality constraint is implemented which keeps the farm power constant. This is usually the case for a tendered wind farm, where the grid connection is a given. The constraint implies that with an increase in the rated power of the turbine, the number of turbines reduces to keep the farm power constant. An area equality constraint is also implemented, which represents a fixed plot of ocean area allocated to the developer to build the wind farm. As a result, the absolute spacing between the turbines depends on the number of turbines that are placed within the given area. These constraints are used for the baseline case as it is assumed to be the most representative of how current commercial wind farms in recent years have been tendered (Rijkswaterstaat, 2021). It should be noted that a sensitivity study to these constraints is also carried out and is presented in Section 4.3.

The LCoE of the wind farm is given by Eq. (2) where $L$ is the operating lifetime of the wind farm, $n$ is the year number, and $r$ is the real discount rate. The numerator contains the Capital Expenditures ($C_{\text{CAPEX}}$) that are paid off initially, the summation of all the annual actualized Operation and Maintenance Expenditures ($C_{\text{OPEX}}$), and the costs to decommission the entire wind farm at the end of its lifetime ($C_{\text{DECOM}}$). The denominator contains the summation of the actualized AEP values.

$$\text{LCoE} = \frac{C_{\text{CAPEX}} + \sum_{n=1}^{L} \frac{C_{\text{OPEX}}}{(1+r)^n} + \frac{C_{\text{DECOM}}}{(1+r)^L}}{\sum_{n=1}^{L} \frac{AEP}{(1+r)^n}} \tag{2}$$

The design space w.r.t. the two design variables, rated power ($P$) and rotor diameter ($D$), is shown in Fig. 2. The entire wind farm level framework will be run for these discrete sets of points. On the secondary y-axis, it can be seen that to keep the farm power constant, the number of turbines reduces as the rated power of the turbine increases. To evaluate a property of interest for any given combination of rated power and rotor diameter, a polynomial surface is then fitted to the data at these discrete points.

## 2.4 Models

This section provides a brief overview of all the models in the framework, highlighting the independent input parameters for each model. Some additional modeling details are discussed in Appendix A.

### 2.4.1 Rotor Nacelle Assembly (RNA)

The rotor aerodynamic performance is evaluated using the classic Blade Element Momentum (BEM) theory. The properties of a reference turbine are used as an input to determine the aerodynamic and structural performance and other RNA properties. The values of power coefficient ($c_{\text{P}}$) and thrust coefficient ($c_{\text{T}}$) in the partial load region are evaluated using the airfoil distribution,

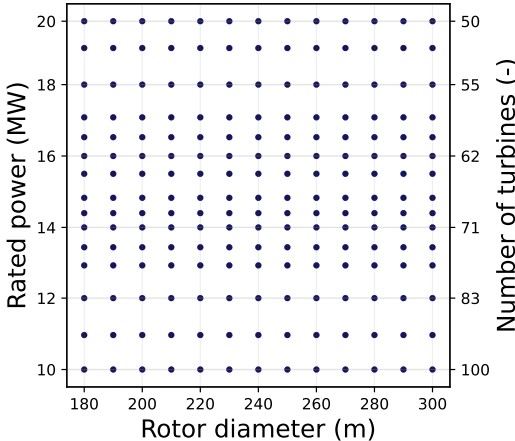

**Figure 2.** Complete design space showing all the combinations of rated power and rotor diameter

the normalized chord and twist profiles, and the tip speed ratio from the reference turbine. It should be noted that since the aerodynamic properties are the same as that of the reference, the resulting peak power and thrust coefficient are the same for all the designs. The rated wind speed of the turbine can then be determined followed by the evaluation of the power curve. The values used for the cut-in wind speed, cut-out wind speed, and the drivetrain efficiency (3 ms$^{-1}$, 5 ms$^{-1}$, and 94.5%, respectively) are the same for all turbine designs.

The rotor mass scaling model uses classical geometric scaling rules with an additional factor, as shown in Eq. (3), where $V_{\text{rated}}$ is the rated wind speed. Both the chord of the blade and the internal layup are scaled linearly with radius. Since the thrust coefficient is the same for all the designs, the additional increase/decrease in thrust is due to a change in the rated wind speed for turbines with a specific power different from that of the reference turbine. The ratio of the rated wind speeds compensates for any additional increase/decrease in the thrust, resulting in the same normalized (with rotor radius) tip deflection as that of the reference.

$$M_{\text{rotor}} = M_{\text{ref}} \cdot \left( \frac{D}{D_{\text{ref}}} \right)^3 \cdot \left( \frac{V_{\text{rated}}}{V_{\text{rated,ref}}} \right)^2 \tag{3}$$

The geometric scaling approach provides a comparison of designs that are conceptually equal. When using scaling coefficients derived from empirical relations, effects from changes in technology, materials, specific power, etc. would be included. This is considered undesirable for this study since it is unknown whether these effects may be extrapolated to a larger scale and what the underlying conceptual changes would be. The downside of geometric scaling is that it typically leads to sub-optimal designs.

To determine the cost of the upscaled rotor ($C_{\text{rotor}}$), a simplified approach as shown in Eq. (4) is used. A weight is given to the scaling of costs with blade mass ($\gamma_{\text{mass}}$) and to the non-mass related costs (1 - $\gamma_{\text{mass}}$), where the non-mass component includes tooling, labor, equipment, etc.

$$C_{\text{rotor}} = \gamma_{\text{mass}} \cdot C_{\text{rotor,ref}} \cdot \left( \frac{M_{\text{rotor}}}{M_{\text{rotor,ref}}} \right) + (1 - \gamma_{\text{mass}}) \cdot C_{\text{rotor,ref}} \cdot \left( \frac{D}{D_{\text{ref}}} \right)^{\alpha} \tag{4}$$

For the baseline case, a $\gamma_{\text{mass}}$ of 0.6 is used, while a scaling exponent, $\alpha$, of 2 for non-mass related aspects is used. These numbers are partially derived from the studies performed by NREL and SANDIA National Laboratories (Bortolotti et al., 2019; Todd Griffith and Johanns, 2013; Griffith and Johanns, 2013).

The components of the nacelle include the bedplate, shafts, yaw system, electrical system, generator, etc. The cost of most components scales with the mass, where the component mass is derived using the DrivetrainSE model (NREL, 2015). The cost of the generator ($C_{\text{gen}}$) also scales up with the mass ($M_{\text{gen}}$), where the mass is proportional to the rated torque of the turbine ($\tau_{\text{rated}}$), as shown in Eq. (5).

$\quad C_{\text{gen}} \propto M_{\text{gen}} \propto \tau_{\text{rated}} \tag{5}$

### 2.4.2  Layout

The layout module generates the wind farm layout by arranging the individual turbines based on a pre-defined arrangement. The dependencies of the layout are shown in Eq. (6), where the turbine coordinates $(x_{\text{i}}, y_{\text{i}})$ are determined by the farm area constraint ($A_{\text{C}}$), number of turbines in the farm ($N_{\text{T}}$), and the orientation of the entire layout ($\theta_{\text{L}}$) governed by the dominant
wind directions.

$$x_{\text{i}}, y_{\text{i}} = f(A_{\text{C}}, N_{\text{T}}, \theta_{\text{L}}) \tag{6}$$

In this study, for a given number of turbines, a layout closest to the nearest possible square arrangement is used, with residual turbines added in an incomplete row. Such a setup avoids boundary effects due to irregular layouts and ensures a fair evaluation of wake losses when comparing different turbine designs.

### 2.4.3  Annual Energy Production (AEP)

The overall AEP of the farm depends on several factors, as shown in Eq. (7). The Bastankhah Gaussian model (Bastankhah and Porté-Agel, 2014) along with the squared sum model is used to estimate wind speed deficits and the wake superposition, respectively. The implementation from the PyWake library of Pedersen et al. (2019) is used as is in the framework.

$$\text{AEP} = f(x_{\text{i}}, \, y_{\text{i}}, \, w_{\text{s}}, \, w_{\text{d}}, \, H_{\text{hub}}, \, D_{\text{rotor}}, \, c_{\text{T}} \text{ LUT}, \, P_{\text{turbine}} \text{ LUT}) \tag{7}$$

The wind speed and wind direction vectors are represented as $w_{\text{s}}$ and $w_{\text{d}}$, respectively. The thrust table of the turbine ($c_{\text{T}}$ LUT) along with the wind speed, wind direction, and turbine coordinates is used to determine the wind speed deficit at each turbine.

The power table ($P_{\text{turbine}}$ LUT) is then used to calculate the power at each turbine, summing up to give the instantaneous farm power. The summation of these instantaneous farm power values over one year results in the overall AEP of the farm.

### 2.4.4 Support structure

The sizing module used for the design of monopiles is based on the work of Zaaijer (2013). The tower top diameter is determined by the yaw bearing diameter while the tower length depends on the hub height ($H_{\text{hub}}$), which is scaled with the rotor diameter, as shown in Eq. (8).

$$H_{\text{hub}} \propto D_{\text{rotor}} \tag{8}$$

The length of the transition piece is fixed while the monopile length is the sum of the evaluated monopile penetration depth, water depth, and the overlap with the transition piece. The aerodynamic and hydrodynamic loads and moments are calculated using the site characteristics and turbine data. The wind and wave loading is calculated for normal operation and also for extreme conditions with a 1-year and 50-year occurrence period. Additional safety factors are introduced for ultimate loads and to compensate for fatigue and buckling. The geometry for the tower and foundation is then determined by a root-finding algorithm that equates the calculated stresses to the permissible values. The rocks for scour protection are also sized by the model. The cost model uses empirical cost factors along with the obtained volume and mass values of the structure. Some of the dependencies of the model are shown in Eq. (9). The mass of the tower and foundation of a given turbine ($M_{\text{support,i}}$) depends on the local turbulence intensity ($\text{TI}_i$), the rotor diameter, the maximum thrust on the rotor ($T_{\text{rotor}}$), the mass of the RNA ($M_{\text{RNA}}$), yaw bearing diameter that sets the tower top diameter ($D_{\text{yaw}}$), nacelle frontal area ($A_{\text{nacelle}}$) and its coefficient of drag ($c_{\text{D,nacelle}}$) to calculate the drag forces, and various site parameters ($\lambda_{\text{site}}$). The site parameters include 50-year and 1-year extreme significant wave heights, storm surge, soil sieve size, wave friction angle, etc.

$$M_{\text{support,i}} = f(\text{TI}_i,\ D,\ T_{\text{rotor}},\ M_{\text{RNA}},\ D_{\text{yaw}},\ c_{\text{D,nacelle}},\ A_{\text{nacelle}},\ \lambda_{\text{site}}) \tag{9}$$

### 2.4.5 Electrical system

The model for the electrical system returns the cost of cabling and substations. The length of the export cable is given by the distance between the substation and the grid, taken as an input from the user ($D_{\text{grid}}$). The infield cable length is calculated using the Esau-Williams heuristic module, which results in a branched topology, as implemented by Sanchez Perez Moreno (2019). For the cost of the export cable, a reference mass per unit length (for a 220 kV cable delivering 1 GW) and a reference cost for the same are used as a variable model parameter. For the infield cable, the rated current of the turbine ($I_{\text{turbine}}$) and the length of each string, along with the different cable types (as mentioned in Section 2.5) are used. The substation costs are scaled linearly with farm power w.r.t. the reference costs of a 1 GW offshore farm (BVG Associates, 2019). The cost dependencies of the different components are shown in Eq. (10).

$$C_{\text{export}} = f(P_{\text{farm}}, D_{\text{grid}}) \tag{10a}$$

$$C_{\text{infield}} = f(L_{\text{infield}}, I_{\text{turbine}}) \tag{10b}$$

$$C_{\text{substation}} = f(P_{\text{farm}}) \tag{10c}$$

### 2.4.6 Installation

The installation cost model takes the vessel data, presented in Table 1, as an input to calculate the installation costs of the foundations, turbines, and electrical system. The dependencies are shown in Eq. (11). The cost of installation for the foundation ($C_{\text{installation,foundation}}$) and the turbine ($C_{\text{installation,turbine}}$) are functions of the rotor diameter, as the vessel day rates are assumed to scale linearly with the diameter. This is an approximation made to account for the growing vessel sizes with larger turbines and foundations. The costs largely depend on the number of turbines ($N_{\text{T}}$) or, equivalently, the number of foundations

to be installed. The installation time for the foundation is assumed to be constant whereas, for the turbine, it depends on the installation strategy used. The turbine installation strategy modeled is the one in which the tower is installed first. This is followed by the nacelle, in a bunny-ear configuration of two blades before installing the third blade (Kaiser and Snyder, 2012). Although this method is not used for current-day turbine sizes, the model for this method captures the main dependencies of installation costs on turbine scale parameters. The absolute values of the model should be interpreted with care. $H_{\text{hub}}$ along

with the number of lifts decides the total lifting time, which is then added to a fixed installation time for each turbine. The distance of the site from the nearest harbor ($D_{\text{harbor}}$) determines the travel time for the installation vessel. The electrical installation costs ($C_{\text{installation,electrical}}$) include the costs incurred to install the infield cables, export cables, and substations. The time taken to install the cables depends on the laying and burial rate and the length of the cables ($L_{\text{infield}}$ for the array cables and $L_{\text{export}}$ for the export cable). The time taken to install one substation is fixed, while the number of substations

depends on the farm power $P_{\text{farm}}$.

$$C_{\text{installation,foundation}} = f(D, N_{\text{T}}) \tag{11a}$$

$$C_{\text{installation,turbine}} = f(D, H_{\text{hub}}, N_{\text{T}}, D_{\text{harbor}}) \tag{11b}$$

$$C_{\text{installation,electrical}} = f(L_{\text{infield}}, L_{\text{export}}, P_{\text{farm}}) \tag{11c}$$

### 2.4.7 Operations & Maintenance (O&M)

The operational costs include insurance, logistics, training, etc., and maintenance costs include preventive and corrective maintenance for the turbine and BoS. The overall O&M costs are a function of several variables, as shown in Eq. (12). The vessel day-rates are scaled linearly with $D$. The failure rates per turbine ($F$) and the number of turbines ($N_{\text{T}}$) determine the number of maintenance trips ($N_{\text{trips}}$) to be made, while the cost of the infield cables ($C_{\text{infield}}$) and RNA ($C_{\text{RNA}}$) are used to determine the costs of major replacements. $D_{\text{harbor}}$ is used to calculate the travel time of the vessels.

$$C_{\text{OPEX}} = f(D, C_{\text{RNA}}, C_{\text{infield}}, D_{\text{harbor}}, N_{\text{T}}, F) \tag{12}$$

The total O&M costs are given by a summation of the operational costs ($C_{\text{operations}}$), vessel costs ($C_{\text{vessel}}$), spare part costs ($C_{\text{sp}}$), and salaries paid to the technicians ($C_{\text{technicians}}$), as shown in Eq. (13).

$$C_{\text{OPEX}} = C_{\text{operations}} + C_{\text{vessel}} + C_{\text{sp}} + C_{\text{technicians}} \tag{13}$$

The operational costs are fixed costs incurred by the developer every year. The type of maintenance (preventive or corrective), the failure rates and the number of turbines decide the type of vessel to be deployed, the number of maintenance trips per vessel type, and spare part costs. The total time spent by the vessel for performing repairs (including the transit time), multiplied by the day-rate of the respective vessel type, determines the total vessel costs. As shown later in Table 2, the spare part cost for RNA-related repairs is expressed as a fraction of the RNA costs while the infield cable replacement costs are expressed as a fraction of the total infield cable costs.

## 2.4.8 Other costs

Other costs include 'other turbine costs', 'other costs for installation and commissioning', 'project development and management costs', and 'decommissioning costs'. The other costs related to the turbine ($C_{\text{other,turbine}}$) include, among others, turbine profit margins and warranty, and take up roughly 30% of the overall turbine CAPEX. The other costs related to the farm installation and commissioning ($C_{\text{other,farm}}$) include insurance, contingency, etc., and take up about 10% of the overall farm CAPEX. Costs related to project development and management ($C_{\text{dev}}$) include various surveys, resource assessments, and engineering consultancy, to name a few, and take up 5% of the overall farm CAPEX (BVG Associates, 2019). The decommissioning costs ($C_{\text{DECOM}}$) involve the removal and disposal of the turbine, foundation, cables, etc. A summary of all the 'other costs' is shown in Eq. (14).

$$C_{\text{other,turbine}} = 0.3 \cdot C_{\text{CAPEX,turbine}} \tag{14a}$$

$$C_{\text{other,farm}} = 0.1 \cdot C_{\text{CAPEX,farm}} \tag{14b}$$

$$C_{\text{dev}} = 0.05 \cdot C_{\text{CAPEX,farm}} \tag{14c}$$

$$C_{\text{DECOM}} = f(L_{\text{cables}}, N_{\text{T}}, M_{\text{RNA}}, H_{\text{hub}}) \tag{14d}$$

## 2.5 Model parameters

To run the MDAO framework as an analysis block, several model parameters are required, as listed below.

1. **Turbine parameters:** The International Energy Agency (IEA) 15 MW turbine (Gaertner et al., 2020) is used as a reference for scaling various properties of the turbine being designed. To scale the aerodynamic properties and scale the structural properties of a given turbine design, the airfoil properties, hub height ($H_{hub}$), tip speed ratio ($\lambda$), chord ($c_r$), twist ($\theta_r$), and mass distribution ($m_r$) of the reference turbine are used. Also, the mass of several components in the nacelle, like bedplate ($M_{bedplate}$) and generator ($M_{gen}$), are scaled from the reference-turbine values, while designing the new turbine.

2. **Cable types:** A list of different cables, each defined by their cross-sectional area ($A_{cable}$), current carrying capacity ($I_{cable}$), and cost per meter ($C_{cable}$), is used as an input while making a selection for the array cables. The cable type selected depends on the rated current of the turbine and the number of turbines in a string. The cost per unit length of different cables is shown in Appendix A.

3. **Vessel data:** The vessel data[1], as shown in Table 1, is used to calculate the installation and O&M costs of the wind farm. The cost data used is based on Dinwoodie et al. (2015), Smart et al. (2016), BVG Associates (2019), Shields et al. (2021), and Mangat et al. (2022).

**Table 1.** Vessel data used for installation and O&M cost modelling.

| Vessel type | Purpose | Day-rate (€) | Transit speed (kmhr$^{-1}$) | Mobilization costs (€) |
|---|---|---|---|---|
| WTIV | Installation foundation, turbine, O&M | 200000 | 10 | 500000 |
| HLV | Installation: Substation | 500000 | 7 | 500000 |
| CLV | Installation: Cable lay | 110000 | 6 | 550000 |
| CBV | Installation: Cable burrial | 140000 | 6 | 550000 |
| CTV | Crew transfer | 3000 | 40 | - |
| DSV | O&M: Scour repair | 75000 | 6 | 225000 |

The cable laying rate of the CLV and the burial rate of CBV are also used as inputs.

4. **Failure rates:** The O&M cost model uses the expected number of minor and major failures for the turbine and BoS as inputs to determine the number of trips required by the respective vessel. The failure rates and spare part costs are derived from Dinwoodie et al. (2015), Shields et al. (2021), Smart et al. (2016) and Mangat et al. (2022), as shown in Table 2, where $C_{RNA}$ represents the cost of a single RNA, $E$ represents the total number of failure events in a year ($F \cdot N_T$), and $C_{infield}$ represents the total cost of infield cables.

---

[1]WTIV - Wind Turbine Installation Vessel, HLV - Heavy Lift Vessel, CLV - Cable Laying vessel, CBV - Cable Burial Vessel, CTV - Crew Transfer Vessel, DSV - Diving Support Vessel

**Table 2.** Failure types and their respective failure rates, repair times, vessel type required, and spare part costs.

| Failure type | $F$ (Expected no. of failures/turbine) | Repair time (h) | Vessel type | Spare part cost |
|---|---|---|---|---|
| Minor repair | 3 | 7.5 | CTV | $0.001{\cdot}C_{\mathrm{RNA}} \cdot E$ |
| Major repair | 0.3 | 22 | CTV | $0.005{\cdot}C_{\mathrm{RNA}} \cdot E$ |
| Major replacement | 0.08 | 34 | WTIV | $0.1{\cdot}C_{\mathrm{RNA}} \cdot E$ |
| Scour repair | 0.023 | 8 | DSV | - |
| Cable replacement | 0.0004 | 32 | CLV | $0.0025{\cdot}C_{\mathrm{infield}}$ |

## 2.6 Case study

This section discusses the case study analyzed for the formulated problem. A hypothetical site and wind farm in the North
Sea are considered. The site parameters and the farm orientation define the case study. The wind rose for the hypothetical
site, shown in Fig. 3a, uses ERA5 Reanalysis data for a location near the Borselle wind farm in the North Sea (Hersbach
et al., 2018). It can be seen that the South-West direction has the highest probability of all wind speed occurrences. This is the
reason why the reference farm layout is oriented towards the South-West direction. Since there is a farm-power and a farm-area
constraint, the number of turbines in the farm depends on the rated power of the turbine, and the normalized spacing depends
on the rotor diameter. The sample layout in Fig. 3b illustrates the wind speed deficits for the 15 MW reference turbine with
67 turbines and an approximate farm power of 1 GW. It can be seen how the turbines are first arranged in a square grid of 64
turbines and the remaining 3 turbines are added along a new column.

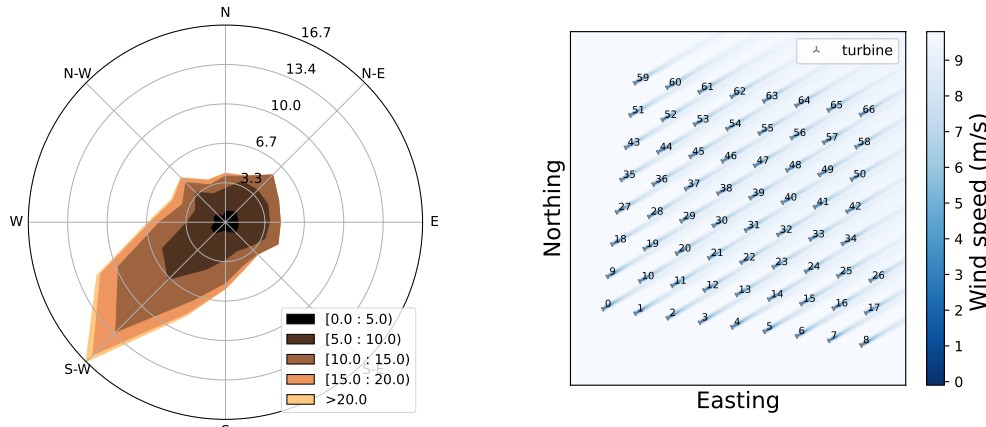

**Figure 3.** (a) Directional wind speeds and probabilities for the hypothetical site. (b) Farm layout for a 15 MW turbine and 1 GW of farm
power.

Some other case-defining parameters, like the distance to grid, water depth, etc., representative of tendered wind farms in the North Sea (Rijksdienst voor Ondernemend Nederland (RVO), 2018), are listed in Table 3. The mean wind speed mentioned is at 100 m and is always projected, using the power law, to the hub height of the turbine design being analyzed.

**Table 3.** Case study parameters

| Parameter | Value | Unit |
|---|---|---|
| Distance to grid | 60 | km |
| Distance to harbor | 40 | km |
| Water depth | 30 | m |
| Mean wind speed at 100 m | 9.4 | ms$^{-1}$ |
| Maximum wave height (50 year) | 5 | m |
| Wind farm lifetime | 25 | years |

## 3    Baseline results

This section discusses the results for the defined baseline case and shows the effect of the two design variables on various farm-level parameters. First, the results with similar specific power designs are presented for a better understanding of general upscaling trends often observed in the industry, followed by the results for the entire design space. The latter gives an overall idea about the changes in turbine design and the specific power of the optimum designs and also discusses the possibility of a global optimum.

### 3.1    Similar specific power designs

The LCoE cost breakdown for a 10 MW turbine and a 20 MW turbine with a similar specific power is shown in Fig. 4. In both cases, the farm power and area are kept constant. It can be seen how the share of turbine costs (rotor, nacelle, tower) goes up as the turbine is upscaled. However, the O&M costs drop, mainly because of a lower number of turbines (for a larger turbine rating). The same reason also accounts for the reduction in turbine and foundation installation cost share. The absolute costs of most of the electrical system (export cable and substation) are constant, as the farm power is unchanged. However, the array cable costs go up for the upscaled turbine, due to an increase in the array cable cost. This can be attributed to a higher current flowing through each string of five turbines.

The effect of upscaling turbines (with the same specific power) on various farm-level parameters is shown in Fig. 5a and Fig. 5b. It can be seen that the overall costs of the turbine and the support structure go up with upscaling. This indicates a non-linear increase in the absolute costs per support structure. The cost of the RNA is dominated by the increase in rotor and generator costs, while the increase in support structure (tower and foundation) costs can be mainly attributed to higher hub heights, higher mass of the RNA, and the increase in thrust. As the export cable and substation costs are fixed due to a fixed farm power, the increase in infield cable costs (due to a higher current carried through the cable) results in an increasing trend

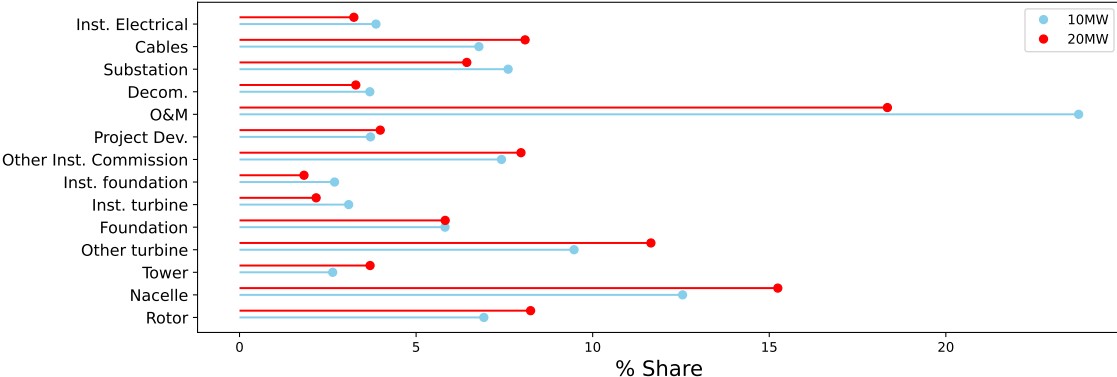

**Figure 4.** LCoE cost breakdown of a 10 MW turbine and a 20 MW turbine, both with a specific power of 350 $\text{Wm}^{-2}$.

of electrical costs. It can also be seen how upscaling decreases the installation and O&M costs. This can be largely attributed
to the decrease in the number of turbines, as the rated power of the turbine increases (for the same farm power). This decrease
in the number of turbines results in a lower number of failure events, reduced vessel time required offshore, and hence, lower
vessel costs. The increase in AEP can be attributed to two main reasons. First, upscaling the rated power results in a lower
number of turbines in the farm (for the same farm power), resulting in lower wake losses. Second, upscaling the rotor diameter
leads to a linear increase in the hub height, resulting in higher wind speeds.

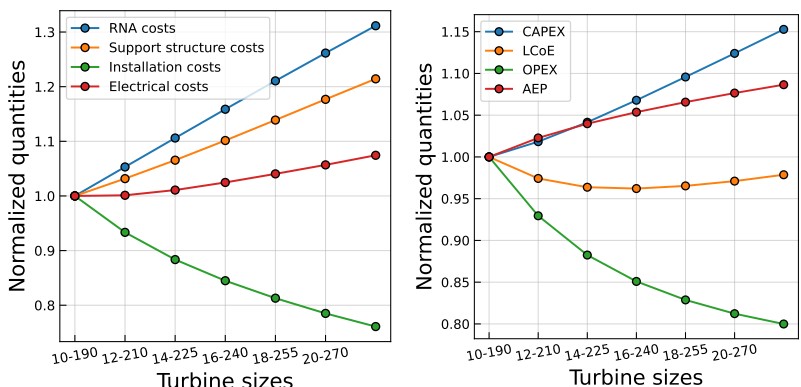

**Figure 5.** (a) Change in components of capital expenditures (normalized) w.r.t. upscaling of turbines with the same specific power. (b) Change in crucial farm-level quantities (normalized) w.r.t. upscaling of turbines with the same specific power.

It can be seen, in general, how upscaling results in a decrease in the O&M costs, an increase in the overall capital expenditures, and an increase in AEP, all of which significantly contribute to the LCoE. However, the trade-offs result in the possibility of an optimum w.r.t. LCoE, as shown in Fig. 5b.

## 3.2 Complete design space

The results for the entire design space explored are presented in this section. As all possible combinations of power and diameter are considered, the effect on various farm-level parameters can also be observed for designs with different specific powers. The magnitude and direction of the gradients of the elements in LCoE (Eq. (2)) at each design point can offer some interesting insights. The cost (capital expenditures, O&M, and decommissioning), AEP, and the LCoE gradients at each evaluation point are shown in Fig. 6 for the baseline case.

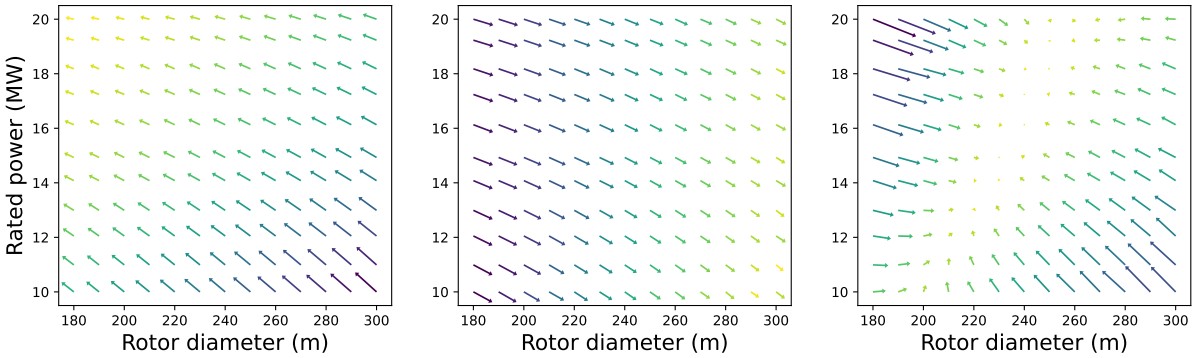

**Figure 6.** Cost, AEP, and LCoE gradients for the baseline problem formulation (from left to right).

It can be seen how the cost gradients always point toward a higher rating and lower diameter as the direction for the steepest descent, while all AEP gradients point towards lower ratings and larger diameters as the direction for the steepest ascent. The magnitude of the cost gradients changes rapidly with a rated power change, while the magnitude of the AEP gradients changes rapidly with a rotor diameter change. From the LCoE gradients, which are a resultant of the cost and AEP gradients, it is evident that there exists a region where the gradient magnitudes are close to zero. This is indicative of a global optimum.

The global optimum for the formulated problem and the defined case is marked in Fig. 7a. In the colormap, each contour line represents a 1% change in the LCoE. This information is useful for a designer, as it indicates the increase in LCoE for a design that deviates from the optimum. It can be seen that a 1% change in the LCoE encompasses a large range of turbine designs, indicating that a deviation in the optimum does not necessarily correspond to a large deviation in the LCoE. This, however, is subject to uncertainties in the models. The plot also includes the largest turbines announced by some turbine manufacturers across the globe, where a majority is already within a 1% LCoE range from the baseline optimum obtained in this study. The

equal specific power lines show that the baseline optimum and the commercial turbines have a specific power range of 300-400 Wm$^{-2}$. It can also be seen how the LCoE variations are smaller when moving along a fixed specific power line compared to the direction of changing specific power.

      At times, the industry has been constrained by the blade length and it can be useful to know the optimum generator rating

for that given rotor diameter. Fig. 7b shows the optimum rated power for a given rotor diameter and the optimum diameter for a given rating. While the 'optimum rated power' line follows a near-constant specific power trend, the specific power

of the 'optimum rotor diameter' follows an increasing trend (300 Wm$^{-2}$ for the 10 MW turbine to 425 Wm$^{-2}$ for the 20 MW turbine). For a farm with a higher-rated-power turbine and fewer turbines, the share of O&M and installation is already relatively low, making the optimum more sensitive to turbine costs. As a result, the specific power increases with an increase in the rated power of the turbine. The cross-over point corresponds with the global optimum. Fig. 7c shows the variation of LCoE as a function of the specific power where the LCoE changes rapidly beyond a certain range of specific powers.

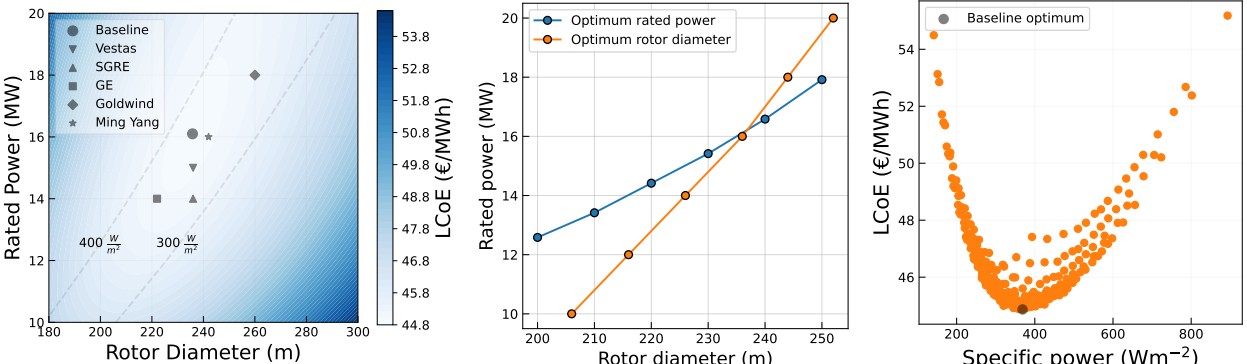

**Figure 7.** (a) Response map of LCoE w.r.t. the two design variables. (b) Optimum rated power and rotor diameter per constrained diameter and rated power, respectively. (c) LCoE plotted against the specific power

To understand several trade-offs that occur at a wind farm level and how they are affected by turbine design, it is useful to see the gradients of various cost and AEP components at the optimum. The gradient of LCoE w.r.t. the rotor diameter and rated power is shown in Eq. (15) and Eq. (16), respectively. It is expressed in the form of cost and AEP gradients along with their weights.

$$\frac{\partial \text{LCoE}}{\partial D} = \frac{1}{\text{AEP}^2}\left(\text{AEP}\cdot\frac{\partial C}{\partial D} - C\cdot\frac{\partial \text{AEP}}{\partial D}\right) = A\cdot\frac{\partial C}{\partial D} - B\cdot\frac{\partial \text{AEP}}{\partial D} \tag{15}$$

$$\frac{\partial \text{LCoE}}{\partial P} = \frac{1}{\text{AEP}^2}\left(\text{AEP}\cdot\frac{\partial C}{\partial P} - C\cdot\frac{\partial \text{AEP}}{\partial P}\right) = A\cdot\frac{\partial C}{\partial P} - B\cdot\frac{\partial \text{AEP}}{\partial P} \tag{16}$$

The weights A and B are shown in Eq. (17).

$$A = \frac{1}{\text{AEP}} \quad \text{and} \quad B = \frac{C}{\text{AEP}^2} \tag{17}$$

The overall cost gradient is simply a summation of the gradients of various costs like turbine, O&M, installation, and other farm costs, as shown, only w.r.t. the rotor diameter, in Eq. (18). The gradients w.r.t. the rated power can be similarly obtained.

$$\frac{\partial C}{\partial D} = \frac{\partial}{\partial D}(C_{\text{turbine}} + C_{\text{other}} + C_{\text{support}} + C_{\text{installation}} + C_{\text{OPEX}} + C_{\text{electrical}}) \tag{18}$$

The net AEP is a function of the wake losses ($\lambda_{\text{wake}}$) and the gross AEP (without wake losses). The gradient of net AEP, hence, can be expressed as the summation of gradients for gross AEP and wake losses, as shown, only w.r.t. the rotor diameter, in Eq. (19). The gradients w.r.t. the rated power can be similarly obtained.

$$\frac{\partial \text{AEP}}{\partial D} = \frac{\partial}{\partial D}\left(\text{AEP}_{\text{gross}} \cdot (1 - \lambda_{\text{wake}})\right) = (1 - \lambda_{\text{wake}}) \cdot \frac{\partial \text{AEP}_{\text{gross}}}{\partial D} - \text{AEP}_{\text{gross}} \cdot \frac{\partial \lambda_{\text{wake}}}{\partial D} \tag{19}$$

Gradients of costs and AEP components that include the weights A and B are indicated with an accent. The gradients at the LCoE optimum are shown in Fig. 8, where the cost gradients are negative and point in the direction of decreasing costs. For the cost gradients, it can be seen how RNA and 'other' costs have the highest magnitudes and have a higher dependence on the rotor diameter. The RNA costs decrease mainly with a decrease in the rotor diameter due to the lower blade mass. Its gradient points towards increasing the rating, as a higher rated power would result in a lower number of turbines in the farm (due to the farm power constraint), decreasing the overall cost of the turbines without a significant increase in the cost per turbine. The support structure costs show a similar behaviour but have a lower magnitude. The O&M and installation costs exhibit a higher dependence on the rated power of the turbine compared to the rotor diameter. This can be attributed to the reduced number of turbines with upscaling, leading to fewer installation trips or a low number of major replacements. This results in low vessel costs. Their gradients also point towards a lower diameter, because a reduction in the rotor diameter reduces the required vessel size and hence, the vessel costs. The costs of the export cable and substation are held constant due to the equality constraint on the farm power. However, the array cable costs change and the infield cable topology depends on the number of turbines in the farm. Owing to the farm area equality constraint, a low number of turbines results in turbines being spread apart. The absolute distance between the turbines is only a function of the number of turbines in the farm and does not depend on the rotor diameter. It can be seen that the electrical-cost gradient points toward lower rated power. This is because a lower rated power results in lower current flowing through a string of 5 turbines, hence reducing array cable costs.

The gross AEP increases with an increase in rotor diameter and a decrease in the rated power. A higher rotor diameter gives a larger swept area and higher power (and hence AEP) for the same wind speed. Similarly, a decrease in the rated power, for the same rotor diameter, results in high-capacity-factor (or low-specific-power) turbines. Turbines with a higher capacity factor result in a higher gross AEP than lower capacity factor designs, irrespective of the rated power of the turbine, as the total farm power is constant in both cases. The wake losses point towards a higher rating and a lower rotor diameter. A higher rating results in a lower number of turbines resulting in a lower overall wind speed deficit. A lower number of turbines also increases the absolute distance between the turbines as they are placed further apart due to the equality constraint on farm area. As the absolute distance is a function of the number of turbines and is fixed for a given turbine rated power, reducing the rotor diameter results in a higher normalized spacing, again contributing to lower wind speed deficits. It should be noted that turbines with a higher rating and smaller rotors (high specific power) have high wake losses. This is because they have a larger partial load

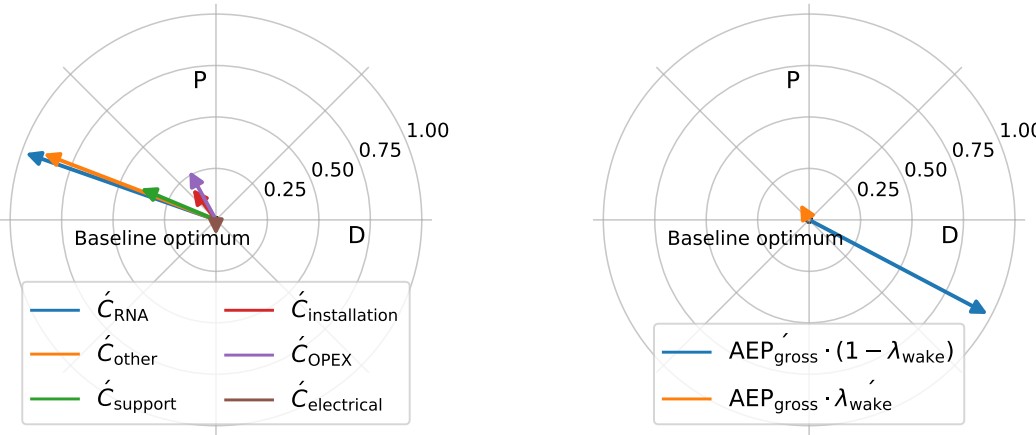

**Figure 8.** Cost and AEP gradient components at the optimum for the baseline case.

region, where the wake losses are the largest. In this region, the turbine operates at high $c_T$ and a reduction in wind speed leads to a decrease in power, unlike in the full load region. However, this effect is overpowered by the reduction in the overall wind
speed deficit caused by a lower number of turbines and larger normalized spacings.

    The possibility of a global optimum indicates that the existing trend of continuous upscaling needs to be closely examined. However, it should be noted that the absolute value of the optimum depends on the current assumptions and future developments w.r.t. technology and costs.

### 3.3   Significance of gradient components

The behaviour of individual contributions to the gradients at the optimum, in a typical wind farm, in terms of both magnitude and direction, is discussed in Section 3.2. The understanding of this behaviour is key in identifying drivers for turbine sizing in a typical wind farm. To understand how certain changes in technology, farm conditions, or specific tendering requirements affect the optimum, one could simply look at how the changes impact the individual gradients and their weightage.

    For instance, a change in the fixed costs, like the costs of export cables or substation, changes only the weightage of the AEP
gradient (see Eq. (17)). This implies that if cables get more expensive, the optimum turbine shifts towards lower specific power turbines due to the stretching of the AEP gradient. Similarly, the removal of the export cable costs reduces the weightage of the AEP gradient, shifting the optimum towards higher specific power turbines. This situation applies, for instance, to the Netherlands, where the transmission system operator provides an offshore electrical connection. Sometimes, the effect on the gradient is more complex, such as for instance a change in blade material. Such a change alters both the magnitude and direction
of the RNA cost gradient, resulting in a shift in the optimum along the direction of constant specific power.

    Such an approach is useful since it shows how drivers that alter mainly the weightage of the gradients (like changes in fixed costs, wind resource, etc.) shift the optimum in the direction of changing specific power, where the impact on LCoE is also

significant (see Fig. 7a). On the other hand, drivers that alter both the direction and magnitude of the gradients (like some technological changes) shift the optimum in the direction of constant specific power, where the impact on LCoE is insignificant (see Fig. 7a). Since the framework uses low-fidelity models, the absolute values of LCoE and optimum designs should not be taken at face value. However, the values match reasonably well with those observed for recently announced turbines and wind farms, adding confidence to the veracity of the results. The analysis of gradient components shows how the framework captures the essential dependencies and how it can be useful in identifying key drivers.

Models that don't include these dependencies might lead to misleading conclusions. This can also be explained by analyzing the gradients. For instance, a model wherein the turbine costs are expressed purely as a function of rated power would assume that the costs increase linearly with the rating. In that case, an increase in the turbine rating from 10 MW to 20 MW would double the costs of an individual turbine while the number of turbines in the farm is reduced to half (due to the farm power constraint). Hence, the total costs of the turbines in the farm remain constant across the entire design space, resulting in the gradients for the RNA costs being zero. As a consequence of this model assumption, the total cost gradient would significantly decrease in magnitude and would now be skewed more in the direction of the rated power. The net resultant of the total cost gradient and the AEP gradient would then significantly push the optimum toward larger ratings and rotors. Practitioners and scientists who focus on LCoE accuracy and fidelity of specific models may overlook the effect that misrepresentation of dependencies may have on gradients in an optimization problem. The insights from this paper may help them make model developments that best match the needs for usage in an MDAO framework.

## 4   Sensitivity study

The results presented so far represent the baseline case for a chosen value for each user input and model parameter. However, to identify the design driving parameters and address the uncertainty in these parameters, a sensitivity study is performed. The parameters chosen are directly influenced by the design variables and are seen to have a significant impact on either the costs or the AEP. Different types of sensitivities are carried out in this study, which can be broadly categorized into the sensitivity w.r.t. the model parameters, design inputs, and problem formulation.

### 4.1   Model parameters

The parameters used in the model are subject to variations either due to uncertainties in estimation or due to differences in future technologies or different economic conditions. The sensitivity w.r.t. the model parameters take into account the variations in the optimum design due to these parameter variations. The choice of rotor diameter and the rated power of the turbine have a large influence on the rotor costs and the O&M costs. Hence, the sensitivity study w.r.t. these two models is performed.

### 4.1.1   Rotor mass and costs model

As the rotor diameter has a direct influence on the rotor costs, and rotor costs take up a noticeable share in the LCoE, a sensitivity is performed w.r.t. the parameters used in rotor mass and cost scaling. The parameters considered for the sensitivity analysis

w.r.t. the rotor and their range of values are listed in Table 4. All the parameters can be found in Eq. (4). The mass scaling
coefficient is used to scale the rotor mass with the rotor diameter and can differ depending on technological developments.
The baseline value used for the reference blade cost is originally scaled from existing data on costs of 90-100 m blades (BVG
Associates (2019)). However, the reference cost per se has uncertainties, and hence, a range of 60% about the same is used.
These parameters vary depending on a change in the material, technological/cost developments, design environment, etc. On
the other hand, the variations in the reference cost of the blade can be attributed to uncertainties in quantifying the same.

**Table 4.** Parameters used for quantifying the sensitivity of the optimum designs w.r.t. the rotor.

| Parameter | Baseline value | Range |
| --- | --- | --- |
| Mass scaling coefficient (Diameter exponent) | 3 | (2,3.5) |
| Mass weightage ($\gamma_{\mathrm{mass}}$) | 0.6 | (0.4,1) |
| Non-mass scaling coefficient ($\alpha$) | 2 | (1.5,4) |
| Normalized cost of reference blade ($C_{\mathrm{rotor,ref}}$) | 1 | (0.7,1.3) |

The spread of global optimum designs for random combinations of the parameters within the ranges given in Table 4 can be
seen in Fig. 9a. The individual effect of the cost-model parameters (where one parameter is varied keeping all other parameters
at their baseline values) can be seen in Fig. 9b. For better readability, the optimum designs corresponding to the lower end of
the parameter variation range are plotted with a larger marker size.

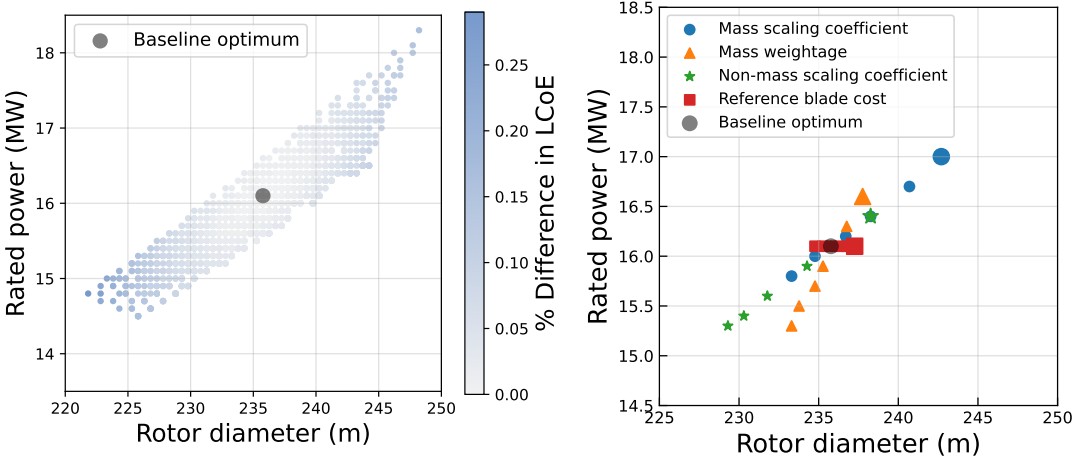

**Figure 9.** (a) Sensitivity of the optimum design to rotor model parameters for random combinations of parameter values. (b) One-at-the-time
variation per parameter (the larger markers correspond to the lower end of the parameter variation range).

A higher value of each of these parameters pushes the global optimum to the left and below the baseline optimum (and
vice versa). A decrease in the mass scaling coefficient results in less than cubic scaling of the mass w.r.t. the diameter, also

resulting in lower rotor costs than the baseline case (for the same diameter value). As a result, the optimum shifts towards larger rotors. An increase in the mass weightage coefficient results in a higher contribution of the material costs to the overall rotor costs. This again pushes the optimum towards smaller rotors. Similarly, an increase in the non-mass scaling coefficient increases the rate at which non-mass-related costs scale w.r.t. the diameter. The result is a shift in optimum towards smaller rotors. The reference cost of the blade, however, does not have a significant impact on the optimum designs. While the other parameters directly influence how the costs scale with the rotor diameter, the reference cost of the blade, as seen in Eq. (4), does not directly influence the cost scaling w.r.t. the diameter. Instead, it only sets the weight of rotor costs, relative to other cost components.

Hence, the optimum designs are observed to be quite sensitive to the mass scaling coefficient, mass weightage, and the non-mass scaling coefficient. It should be noted that although the sensitivity leads to a large variation in the global optimum, a minimal difference in LCoE is observed between the new optimum designs and the baseline optimum.

### 4.1.2 O&M costs model

The O&M costs have the largest share in the LCoE of the farm and the cost model is also quite sensitive to its input parameters. Hence, a sensitivity study is performed w.r.t the parameters shown in Table 5. The fixed costs related to the operations vary depending on the project and location. The failure rate refers to the major replacements in the RNA and is turbine technology dependent, but it also has uncertainties in its estimation. The vessel day rates also depend on the design location and are quite volatile. Also, the scaling of vessel day rates with turbine sizes is uncertain. The vessel mentioned here refers to the WTIV, which is also used for major replacements.

**Table 5.** Parameters used for quantifying the sensitivity of optimum w.r.t. O&M.

| Parameter | Baseline value | Range |
|---|---|---|
| Fixed costs (M €) | 22.5 | (10,30) |
| Failure rate (%) | 8 | (4,12) |
| Vessel day-rate (M €) | 0.2 | (0.1,0.3) |
| Vessel day-rate scaling coefficient | 1 | (0,3) |

Fig. 10a shows the global spread of the optimum designs w.r.t. the variations in the O&M model parameters and inputs. The individual effect of the cost-model parameters (where one parameter is varied keeping all other parameters at their baseline values) can be seen in Fig. 10b. For better readability, the optimum designs corresponding to the lower end of the parameter variation range are plotted with a larger marker size. The fixed costs have an indirect effect on the optimum. A change in the fixed costs affects the weight of the AEP gradient (Eq. (17)). A lower fixed cost reduces the weight of the AEP gradient, pushing the optimum toward the direction of the cost gradient (smaller rotors and larger ratings). The fixed costs don't alter the cost gradients but only the weight of the AEP gradient, increasing/ decreasing its magnitude. Hence, the optimum moves along the direction of the cost/AEP gradient. Except for the fixed costs, changes in any other parameter alter the O&M gradient

magnitude and hence, change both the magnitude and direction of the total cost gradient. The overall vector sum of the changes in the direction of the cost gradient along with the changes in the magnitude of the cost and AEP gradient causes a shift along the constant specific power line. For a low failure rate, the number of trips for major replacements is reduced, resulting in lower vessel and spare part costs. This pushes the optimum designs towards a lower rated power (and higher number of turbines) than that of the baseline one. The vessel day rates affect the overall vessel costs. Like the failure rate, this affects the total repair costs and it thus affects the optimum in the same direction. However, since the vessel costs are only a part of the total repair costs, the difference in the optimum designs is not as significant for the given range of day rates. The vessel day rate scaling coefficient, on the other hand, affects how the day rates scale w.r.t. the rotor diameter. A coefficient of zero results in the same day rate, no matter what the turbine size is, incentivizing upscaling. A high coefficient quickly scales up the day rates with rotor size, resulting in smaller rotors.

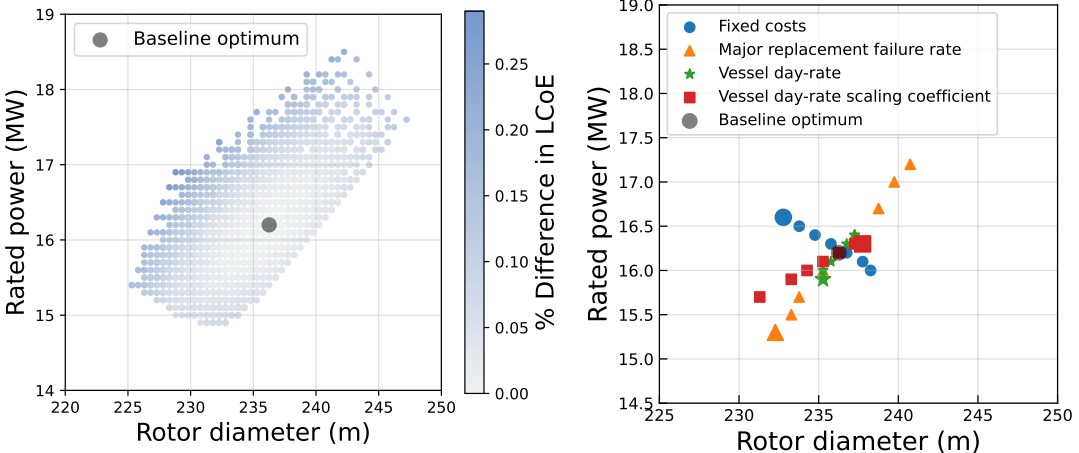

**Figure 10.** (a) Sensitivity of the optimum design to O&M model parameters for random combinations of parameter values. (b) One-at-the-time variation per parameter (the larger markers correspond to the lower end of the parameter variation range).

The fixed cost component of O&M costs, the turbine failure rates, and the scaling of vessel rates with the rotor size are the biggest design drivers, while the uncertainties in the day rate itself do not have a significant impact on the optimum designs. It should be noted that, although the sensitivity leads to a large variation in the global optimum, a minimal difference in LCoE is observed between the new optimum designs and the baseline optimum.

### 4.2 Farm design conditions

The farm design conditions depend on the design environment and are bound to be different for every project. Hence, a sensitivity study is performed w.r.t. several important design conditions like the wind speeds at the site, available farm area, distance to grid, and the total farm power (or grid connection available). The variability range of design conditions is shown in Table 6.

**Table 6.** Parameters used for quantifying the sensitivity of optimum w.r.t. design conditions, where $w_s$ refers to the wind speeds used in the baseline case.

| Parameter | Values |
|---|---|
| Wind speed | $(0.85 \cdot w_s, 0.9 \cdot w_s, 1.1 \cdot w_s)$ |
| Farm area (km$^2$) | (100, 200) |
| Distance to grid (km) | (30, 90) |
| Farm power (MW) | (600, 800, 1200, 1400) |

The effect of the design conditions on the LCoE of these optimums is shown in Fig. 11a, while the variation in the global optimum design w.r.t. the variations in the design conditions can be seen in Fig. 11b. To show the correspondence between Fig. 11a and Fig. 11b, the optimum rotor diameter and rated power values for the lower end of the design input range are mentioned next to their corresponding LCoE values. Compared to the baseline case, it can be seen that a change in the wind speed at the site has the maximum effect on both the LCoE and the optimum design. A low wind speed site pushes for low specific power turbines with larger rotor diameters and lower ratings than the baseline, also resulting in higher LCoE values. This trend is in line with what is seen for turbines in the market and corresponds with typical OEM portfolios. A change in the farm power also has a significant effect on both the LCoE and the optimum designs. Obtaining a larger farm power by increasing the number of turbines (for the same turbine rated power), would increase wake losses, O&M costs, installation costs, electrical system costs, etc. Designs with a higher rated power avoid these effects, and therefore, provide the optimum way to achieve higher farm powers. The available farm area and the distance to grid have a lower impact on the LCoE and, hence, on the optimum design. A low farm area increases the wake losses resulting in a higher LCoE of the farm. To compensate for the high wake losses, the optimum shifts towards higher turbine ratings to reduce the number of turbines in the farm. The distance to the grid only changes the costs of the export cable. Consequently, the effect on the LCoE and the optimum design is minimal. The greatest benefit of tailoring designs to site conditions, in terms of LCoE, is observed for changing wind conditions. In the low wind scenario, opting for low specific power designs, compared to the baseline optimum, resulted in an LCoE reduction of 1.25%.

As the wind regime and the farm power cause the optimum design to shift the most, and along different dimensions, the driving forces behind these shifts are analysed. At the baseline optimum, the gradient of the LCoE is zero, with respect to both changes in diameter and rated power. When the wind regime or farm power changes, the gradient of LCoE at the baseline optimum is no longer zero, and the optimum shifts towards the direction of steepest descent. To identify the contributions of the changes in costs, gross AEP, and wake losses to the direction of the new gradient, the separation of terms according to Eqs. (15) to (19) is shown in Fig. 11c. The weighted cost and AEP gradients for the baseline are equal in size and in opposite directions, in accordance with their vector sum being zero.

For the low wind case, both weighted gradients increase. Although the cost and thus the cost gradient themselves are not affected by the changed wind climate, the weighted cost gradient increases due to the division by the lowered AEP. The AEP gradient is weighted with the square of the AEP, so the effect of the lower AEP on the weighted AEP gradient is much larger.

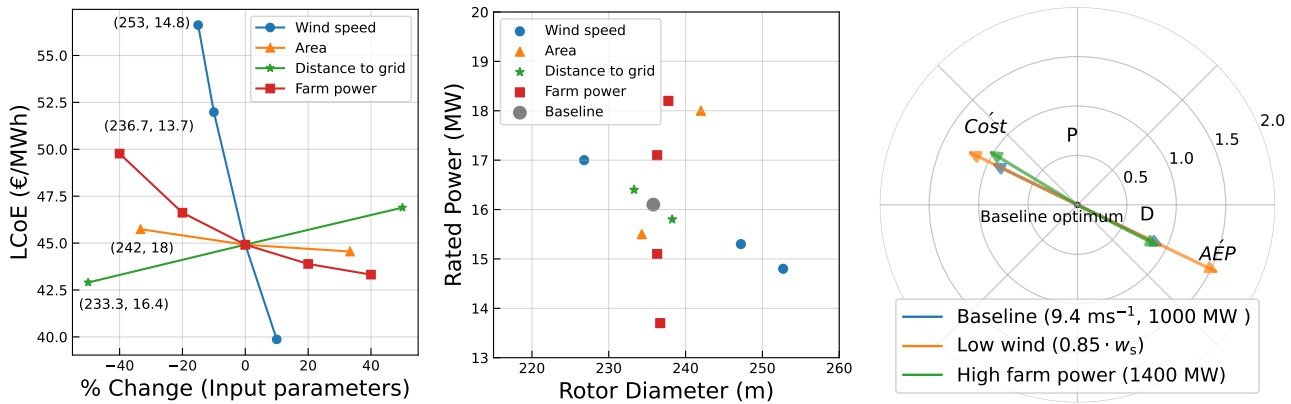

**Figure 11.** (a) Change in LCoE w.r.t. change in design conditions. (b) Change in the optimum w.r.t. change in design conditions. (c) Change in gradients at the baseline optimum for different design conditions.

In addition, the AEP gradient itself will have changed. The sensitivity of AEP to changes in rated power and rotor diameter has changed, since the new probability distribution of wind speeds changes the importance of different parts of the power curve. However, this effect is much smaller. Since the changes in the weighted gradients are dominated by the changes in the weights (shown in Eq. (17)), the two gradient vectors are exactly or mostly aligned with the original vectors. The bigger increase in the weighted AEP gradient pushes the optimum in that direction. Fig. 6 shows that moving in that direction increases the AEP gradient and decreases the costs gradient. The new optimum will be found where this effect compensates for the imbalance that was identified at the baseline point.

A change in farm power (1400 MW) has a more complex effect on the weighted gradients. At the baseline point, the increase in farm power is achieved only by increasing the number of turbines. If AEP and all costs would increase linearly with the number of turbines, the weighted gradients would be the same as for the baseline. The unweighted gradients of costs and AEP would then also increase linearly with the number of turbines, and that would exactly compensate for the linear change in the magnitude of the weights. Therefore, the deviations observed in Fig. 11c are caused by non-linear dependencies. The increase of the number of turbines in the same constraint area leads to a lower average spacing, which in turn leads to two primary non-linear effects: One, the costs increase less than linear, because the total infield cable length grows less than linearly. Two, due to an increase in wake losses, the AEP increases less than linearly. Both have a direct effect on the weights and order of magnitude of the unweighted gradients, which both stretch or compress the baseline weighted gradients along their original orientation. Which non-linearity dominates can only be identified by quantifying them, but according to the graph the differences are small. Both non-linear effects also influence the direction of the unweighted gradients and with that the direction of the weighted gradients. Apparently, for the larger farm, it is a little more favorable to increase the rated power to reduce costs, while the AEP has a similar dependency on the design variables, as that of the baseline. The overall effect of the change in magnitude and direction of the weighted gradients leads to a vector sum that points slightly towards higher

rated powers. This residual gradient for LCoE is far smaller than was the case for the lower wind climate. However, it is now caused partly by differences in the angle between the weighted cost and AEP gradients and not only by a difference in size. This means that at the new optimum point, this difference in angle must be compensated. Fig. 6 reveals that the angles of the gradients change far less rapidly over the domain than their magnitude. This explains why the optimum design for the larger wind farm is at a similar distance from the baseline as the optimum design for lower wind speeds.

Fig. 11b shows that the sensitivity to area is largely aligned with that of the sensitivity to farm power. Likewise, the sensitivity to the distance to grid aligns with that of wind speed. The rationale for each of them resembles the discussion of each of their respective counterparts given above, but for a reversed effect. For changes in the distance to grid, the AEP remains the same and the total costs change. This mainly affects the weight 'B' of the gradients (Eq. (17)) resulting in a slight pull or compression of the weighted AEP gradient. For changes in the area, the AEP and costs both remain nearly the same, with again some effects on wake losses and infield-cable costs. The final effect is due to a combination of small changes in the weights, as well as small changes in the direction of the gradients.

## 4.3    Farm-level constraints

The optimum designs are highly subject to the farm-level constraints themselves. In the baseline case, a fixed farm power and fixed farm area scenario are considered. However, in some cases, there might be a constraint only on the grid connection or only on the available area. Hence, a sensitivity w.r.t. these different constraints is also performed. The problem formulation is varied by removing one of the equality constraints from the baseline case. So other than the baseline case, a problem with a power-only constraint and a problem with an area-only constraint are considered.

### 4.3.1    Fixed farm power

In many scenarios, developers are provided with just a fixed grid connection with no strict limitations on the ocean area. The 'fixed-farm-power-only' constraint represents this scenario where a fixed grid connection of 1 GW is used and a fixed normalized spacing (downwind and crosswind) of 5D is assumed. The overall shift in the optimum is shown in Fig. 12a, while the change in the weighted cost and AEP gradients, plotted at the baseline optimum, is shown in Fig. 12b.

In this constraint formulation, a major change in the behaviour of wake losses and infield cable costs is observed. The change in the magnitude and direction of the cost gradient can be attributed to the differences in the infield cable cost gradient. Unlike the baseline case, the infield-cable costs don't show a significant dependence on the rated power anymore. For a given rotor diameter, the absolute distance between the turbines is fixed. Hence, an increase in rated power reduces the number of turbines, thus decreasing the overall cable length and costs. However, this will also increase the power flowing through a single cable, which increases the costs. These two opposing effects reduce the dependence of infield-cable costs on the rated power. On the other hand, the rotor diameter of the turbine influences the absolute distance between the turbines and hence, the infield cable length. As a result, the infield cost gradient depends more on the rotor diameter than on the rated power, resulting in the overall change in both the direction and magnitude of the net cost gradient, compared to that of the baseline.

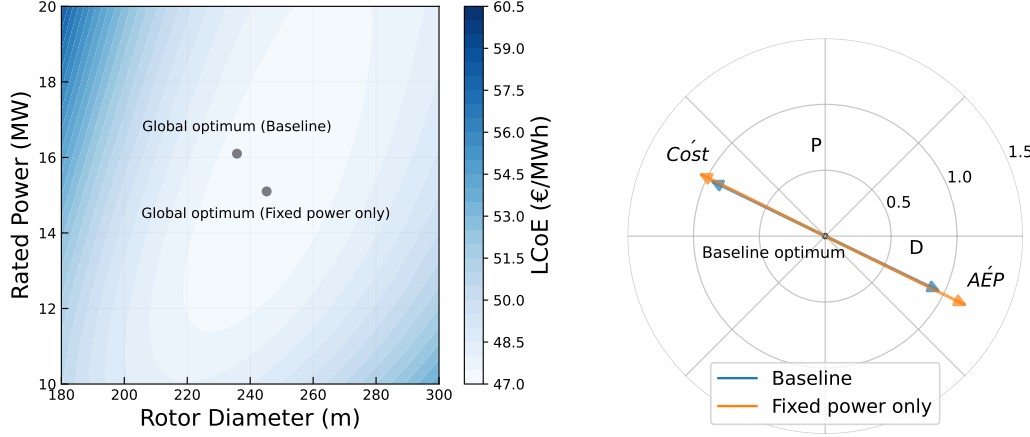

**Figure 12.** (a) Shift in the optimum design for the 'fixed-farm-power-only' case. (b) Differences in gradients for 'fixed-farm-power-only' case plotted at the baseline optimum.

The behavior of wake losses also changes significantly with the change in the constraint formulation. The wind speed deficits depend on the normalized spacing between turbines and the operating thrust coefficient of the upstream turbine. In the baseline case, a change in the turbine power or diameter changed both the normalized spacing and the power/thrust curve of the turbine. However, the wake losses were mainly dominated by the normalized spacing (around 7.5D for the baseline optimum). In the

615 'fixed-farm-power-only' case, the normalized spacing is fixed (at 5D) and doesn't depend on the number of turbines. So a change in the turbine design only changes the power/thrust curve. The wind speed deficit experienced by downstream turbines is highest when the upstream turbines operate in their partial load region, because that is where the thrust coefficient and power gradient are highest. As a result, the net AEP gradient pushes the optimum towards larger rotors and lower ratings (low specific power turbines), which have a steeper power curve and a reduced partial load region. This results in lower wake losses.

Since the change in the wake loss gradient dominates the change in the infield cable cost gradient, the net effect is a push in the resulting optimum towards lower ratings and larger rotors.

### 4.3.2 Fixed farm area

This constraint formulation represents a scenario where the ocean area is limited and developers are mostly area constrained. This applies, for instance, to coastal regions, close to a strong grid connection. The farm area available is considered to be

fixed in this case. For a given rotor diameter and normalized spacing, the area constraint determines the maximum allowable number of turbines, since the farm area will always be used to its full capacity. Also, the turbine rated power has no impact on the number of turbines in the farm. In this analysis, a normalized spacing (downwind and crosswind) of 5D is assumed. The optimum rated power is observed to be much higher than that in the baseline case (Fig. 13a). The change in the weighted cost and AEP gradients, plotted at the baseline optimum, is shown in Fig. 13b. It should be noted that Fig. 13b shows the direction

of the steepest descent/ascent at the baseline optimum, and moving along that direction will lead to another point where the gradient direction will differ, ultimately leading to the global optimum.

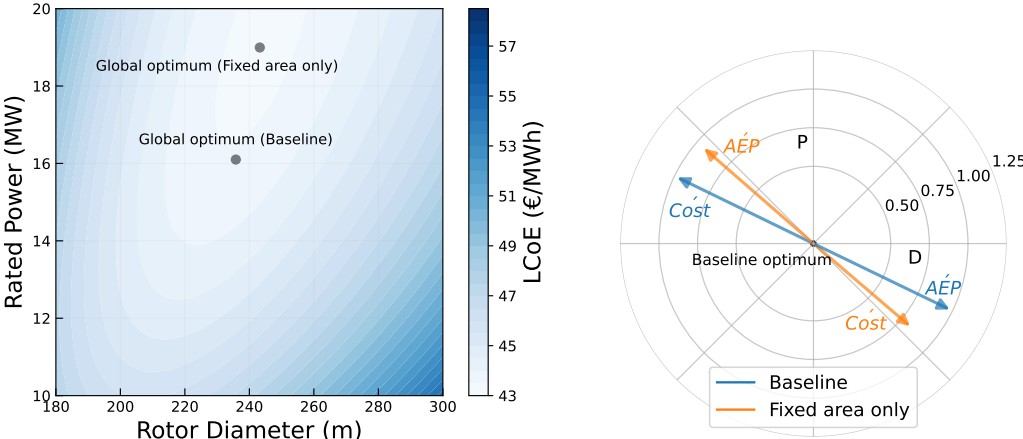

**Figure 13.** (a) Shift in the optimum design for the 'fixed-farm-area-only' case. (b) Differences in gradients for 'fixed-farm-area-only' case plotted at the baseline optimum.

Compared to the baseline (Fig. 8), a completely opposite behavior of both the costs and AEP gradients can be clearly observed. This can be attributed to the net change in the direction and magnitude of the individual cost and AEP components, as explained below:

– A decrease in the rotor diameter decreases the costs of a single turbine but results in more turbines in the farm. These opposing effects reduce the dependence of the total turbine costs for the farm on the rotor diameter. On the other hand, increasing the turbine-rated power increases the RNA costs, and has no impact on the number of turbines. As a result, the RNA cost gradient points toward lower rated powers.

       – As the farm power is now variable and depends on the number of turbines in the farm and the rated power of each
turbine, the costs of the export cable and substations also vary. The infield costs now go down with an increase in the rotor diameter (less turbines in the farm) and lower ratings (less current flowing through the cables). As a result, the overall electrical cost gradient points towards lower rated powers and larger rotors.

       – As the number of turbines is now a function of the rotor diameter, the O&M and installation cost gradients point towards larger rotors. This is because, for the 'fixed-farm-area-only' case, where the area constraint and normalized spacing are
defined, a larger rotor diameter results in a lower number of turbines in the defined area, resulting in lower vessel costs.

       – The gross AEP (without wake losses) of the farm goes up with an increase in the rated power and a decrease in the rotor diameter. This is because a decrease in the rotor diameter allows more turbines in the farm and an increase in the rated

power allows for more power to be produced at higher wind speeds without having any impact on the overall number of turbines in the farm. As a consequence, the gross AEP vector points towards larger rated powers and smaller rotor diameters.

- Similar to the 'fixed-farm-power-only' case, since the normalized spacing is fixed, the wake losses are minimized by reducing the partial load region of the turbine. This is achieved by moving to larger rotors and a lower rated power. Also, larger rotors reduce the number of turbines in the farm, further reducing wake losses.

The effects discussed above explain the difference in the behavior of the cost and AEP gradients. It can be seen that the AEP gradient is larger in magnitude and is also not in line with the cost gradient. This creates a push towards larger ratings and at the new optimum the differences in the directions and magnitude of the weighted cost and AEP gradients are compensated.

## 5   Conclusions

This work explored how turbine scaling affects various farm-level quantities and also identified the design driving parameters. Key insights gained from this research are presented below.

■ The optimum size of 16 MW – 236 m for a typical case is already close to the state-of-the-art observed in the industry. For a farm with a fixed area and a fixed power for the grid connection, reduction of farm costs is attained via high specific power turbines. This is achieved mainly via reducing the rotor diameter. On the other hand, an increase in the AEP is attained via low specific power turbines achieved mainly via increasing the rotor diameter. The apparent trade-off results in an optimum turbine design w.r.t. LCoE. Also, a large change in the turbine design along a constant specific power line results in a small LCoE change. On the other hand, a small change in the turbine design, but in the direction of changing specific power, results in a large LCoE change.

■ Uncertainties in modeling or future technology/cost developments drive the optimum along a fixed specific power line. However, the optimum for the typical case mentioned above is robust to these differences in the model behavior, w.r.t. LCoE. For instance, the variations in the scaling of rotor mass, estimation of failure rates, vessel cost scaling, etc. resulted in an uncertainty range of 10% for the optimum diameter and 20% for the optimum rated power evaluation. However, this uncertainty in the optimum led to a difference of less than 0.5% in the LCoE.

■ Project-specific parameters drive a change mainly in the specific power of the design. The variations in wind conditions and farm power density values were seen to have the largest impact on both the optimum turbine size and the specific power of the design. As an example, for the case presented here, changing the wind regime by 15% mainly affects the rotor diameter of the optimum resulting in a specific power shift from 390 $Wm^{-2}$ to 310 $Wm^{-2}$. On the other hand, varying the farm power density constraint by 40% mainly affects the rated power of the optimum resulting in a specific power shift from 390 $Wm^{-2}$ to 430 $Wm^{-2}$. However, redesigning the turbine for these changes resulted in a maximum benefit of the order of 1-2% w.r.t. LCoE.

- Relative to a typical case, a scenario with only area constraints pushes the optimum towards high specific power designs extracting the maximum energy out of a turbine. A scenario with only farm power constraints pushes the optimum towards low specific power designs reducing the wake losses in the partial load region. Turbine designers can adapt to these changing problem formulations either by downrating the turbine for a 'farm power constrained' scenario or by uprating the turbine for an 'area constrained' scenario.

These findings agree with how technological advancements led to the continued shift of the turbine scale observed in the past decades. Technology and cost developments drive a shift in the optimum scale with limited effect on specific power (second bullet point). The sensitivity of LCoE along lines of equal specific power is low around the optimum, allowing a large range of scales to co-exist during a certain era (first bullet point). The findings also agree with the fairly stable range of specific powers offered in the past at different scales. These portfolios are driven by variations in project-specific conditions (third bullet point). However, while the optimum specific power is fairly sensitive to particular project conditions, turbines of a fairly wide range of scales can perform equally well (first bullet point). The findings of bullet point 4 are less visible in practice and in the literature, but they reveal the effects of farm-level constraints on the optimum specific power and scale that are similar to those of project conditions. Besides the more obvious consequence for farm developers to pick the most suitable turbine for their case, this finding also means that the policies around spatial planning and tender formulation have an impact on the optimum turbine design and performance. An approach as shown in this paper can help quantify those impacts. The findings in this research are obtained using low-fidelity cost models and the IEA 15 MW turbine as the reference design. However, the absolute values of the optimum will likely differ for a different reference turbine as the starting point for scaling, and a future study exploring the sensitivity to different reference designs with higher fidelity cost models is recommended. Nevertheless, the confidence in the use of scaling laws is largest when the scale of the reference turbine and of the global optimum are similar, as is the case here.

The study provides a simplified approach that can be applied to a complex turbine sizing problem in order to generate meaningful insights. The findings of this study help the scientific community to focus future research on the most important aspects and goals. They provide insight into how various model improvements impact both the performance and the optimum turbine size. For instance, consider an improvement in the RNA model leading to an increase in the magnitude of its gradient and a higher dependency on rated power. The consequence of this improvement would be a large shift in the optimum along the constant specific power line, towards larger ratings and rotor diameters, without a significant change in the LCoE for the new optimum. Similar insights can be drawn w.r.t. other model improvements, based on the gradients presented in this study. The findings also show how constraints influence turbine sizing, guiding future studies w.r.t. the optimization problem formulation. The research serves as a stepping stone for the sizing of future reference turbines. However, the marginal change in LCoE for a wide range of designs shows the limited benefit of continuous upscaling. These limited benefits have to be balanced against the technical challenges and risks posed by further upscaling.

*Code availability.* The code along with all the input files and plot scripts is open source and is available at:
https://doi.org/10.5281/zenodo.8380355

## Appendix A: Modeling details

Additional modeling details w.r.t. some disciplines of the wind farm are discussed in this section. The input values used as a reference for the turbine/farm level costs can be found in the open-source code hosted on Zenodo.

### Support structure

The support structure module evaluates several parameters related to the tower and the foundation. The hub height is scaled from the IEA 15MW turbine with the rotor diameter. The tower top diameter ($D_{\text{tower,top}}$) is set to be equal to the diameter of the yaw bearing ($D_{\text{yaw}}$). The length of the transition piece ($L_{\text{tp}}$) depends on the platform height ($H_{\text{platform}}$), which is set based on the maximum wave height with a clearance of 20%, and the base of the transition piece ($H_{\text{tp,base}}$), which is set to be slightly below the water line. Since the hub height is known, the tower length ($L_{\text{tower}}$) can be given by the equation shown below where $y_{\text{tip,platform}}$ is the clearance between the blade tip and the platform.

$$L_{\text{tower}} = H_{\text{hub}} - H_{\text{platform}} - y_{\text{tip,platform}}$$

The combined wind-wave load cases include a regular operation of the turbine at rated wind speed with a maximum wave in a one-year extreme sea state, a parked turbine with reduced gust in a 50-year average wind speed and maximum wave in a 50-year extreme sea state, and a parked turbine with a maximum gust in 50-year average wind speed and reduced wave in 50-year extreme sea state. The diameter of the monopile ($D_{\text{monopile}}$) is evaluated such that the maximum stress due to the combined wind-wave load cases is equal to the yield stress of steel. The diameter of the transition piece ($D_{\text{tp}}$) is set to be 300 mm larger than the diameter of the monopile. The thickness of the monopile is set to be around 1/100 times $D_{\text{monopile}}$. The diameter of the tower base ($D_{\text{tower,base}}$) is set equal to the transition piece diameter. The thickness of the transition piece and the thickness of the tower segments are obtained using Brent's algorithm such that the maximum stresses due to the load are equal to the permissible values. The monopile penetration depth ($L_{\text{monopile,penetration}}$) is set to be 10% larger than the clamping depth obtained using Blum's model for piles that undergo lateral loading. The total monopile length can then be calculated using the equation below, where the total length is a summation of the monopile penetration depth, water depth ($H_{\text{water}}$), height of the transition piece base ($H_{\text{tp,base}}$), and the overlap between the monopile and the transition piece ($L_{\text{monopile,overlap}}$).

$$L_{\text{monopile}} = L_{\text{monopile,penetration}} + H_{\text{water}} + H_{\text{tp,base}} + L_{\text{monopile,overlap}}$$

Once the geometric properties of the tower, transition piece, and monopile are known, the mass and costs are calculated using calibrated cost factors. The module also calculates the properties and costs of scour protection.

### Electrical system

The length of the infield cables is obtained via the Esau-Willaims heuristic module while the distance to the grid determines the length of the export cable. The Cost Factor of the infield cables ($CF_{\text{infield}}$) is a function of the current flowing in the string.

The cost curve used in this study is shown in Fig. A1, where the total string current ($I_{\text{string}}$) is given by the number of turbines in the string and the turbine rated current. For this study, the number of turbines in a string is fixed at 5.

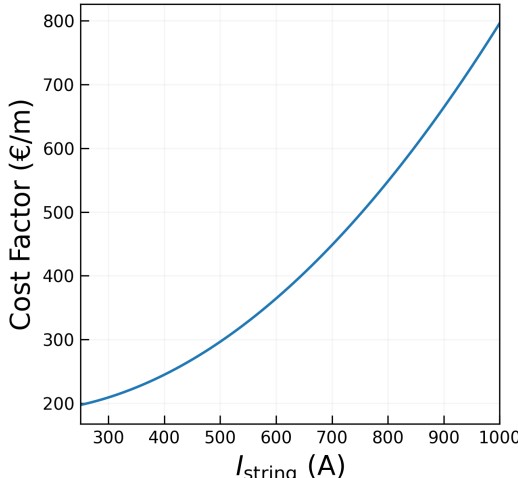

**Figure A1.** Infield cable cost as a function of total current in a string

$$C_{\text{cable,infield}} = \text{CF}_{\text{infield}} \cdot L_{\text{infield}}$$

For the export cable, the mass of the cable ($M_{\text{cable,export}}$) is scaled with the farm power, and the total export cable costs
($C_{\text{cable,export}}$) scale with the mass of the cable and the distance to the grid ($D_{\text{grid}}$). The substation costs ($C_{\text{substation}}$) have a fixed cost component ($C_{\text{fixed}}$) and a variable cost component ($C_{\text{variable}}$) that scales with the farm power. The total costs for the export cable and substation are fixed for any given turbine design in case of a farm power constraint. However, when there is no farm power constraint, these values scale with the farm power as shown in the equations below.

$$M_{\text{cable,export}} = M_{\text{cable,ref}} \cdot N_{\text{T}} \cdot P_{\text{rated}}$$
$$C_{\text{cable,export}} = C_{\text{cable,ref}} \cdot M_{\text{cable}} \cdot D_{\text{grid}}$$
$$C_{\text{substation}} = C_{\text{fixed}} + C_{\text{variable}} \cdot N_{\text{T}} \cdot P_{\text{rated}}$$

The normalized values of the reference are based on the costs of a typical 1 GW offshore wind farm in the UK (BVG Associates, 2019).

### Installation

The summation of the installation costs for the turbine, foundation, and electricals gives the total installation costs. The reference day rate of the Wind Turbine Installation Vessel (WTIV) is scaled with the rotor diameter of the turbine.

The time for one vessel trip to the site includes the time taken to load the RNA ($T_{\text{loading}}$), the time taken to travel to the site ($T_{\text{travel}}$), and the time taken to install the RNA ($T_{\text{install}}$). The total turbine installation time ($T_{\text{installation,turbine}}$) simply depends on the time taken per trip and the total number of trips made by the vessel ($N_{\text{trips}}$). The total time is multiplied by a factor of 1.5 to account for weather delays.

$$T_{\text{installation,turbine}} = N_{\text{trips}} \cdot (T_{\text{loading}} + T_{\text{travel}} + T_{\text{install}}) \cdot 1.5$$

The number of trips depends on the total number of turbines ($N_T$) and the vessel capacity (assumed to be 5 in this study). The total cost for turbine installation is then given by the total vessel costs to install the RNA and the costs to mobilize and demobilize the vessel.

$$C_{\text{installation,turbine}} = C_{\text{day,WTIV}} \cdot T_{\text{installation,turbine}} + C_{\text{mobilization,WTIV}}$$

The cost of installation for the support structure ($C_{\text{installation,support}}$) can be derived in the exact same way, where the total time taken to install the support structure ($T_{\text{installation,support}}$) is used.

For the cable installation, the time taken by the Cable Laying Vessel (CLV) and the Cable Burial Vessel (CBV) depends on the installation rate of the vessels. The total time taken to install the infield and export cable depends on the length of the cable and their installation rates ($r_{\text{installation,infield}}$ and $r_{\text{installation,export}}$, respectively) along with the safety factor for weather delays.

$$T_{\text{installation,infield}} = \frac{L_{\text{infield}}}{r_{\text{installation,infield}}} \cdot 1.5$$

$$T_{\text{installation,export}} = \frac{L_{\text{export}}}{r_{\text{installation,export}}} \cdot 1.5$$

The total cable installation costs are then calculated by multiplying the time taken with the day rate for the vessels along with the mobilizing costs of the vessel, and some extra costs ($C_{\text{extra}}$) for cable pull-in, testing, etc.

$$C_{\text{installation,infield}} = T_{\text{installation,infield}} \cdot (C_{\text{day,CLV}} + C_{\text{day,CBV}})$$

$$C_{\text{installation,export}} = T_{\text{installation,export}} \cdot (C_{\text{day,CLV}} + C_{\text{day,CBV}})$$

$$C_{\text{installation,cables}} = C_{\text{installation,infield}} + C_{\text{installation,export}} + C_{\text{mobilization,CLV}} + C_{\text{mobilization,CBV}} + C_{\text{extra}}$$

Finally, the cost to install the substations is given by the cost of installing the onshore substation ($C_{\text{installation,onshore}-\text{substation}}$), time taken to install the offshore substation ($T_{\text{installation,offshore}-\text{substation}}$), the day rates of the heavy lift vessel ($C_{\text{day,HLV}}$) and the cost to mobilize them ($C_{\text{mobilization,HLV}}$).

$$C_{\text{installation,substation}} = T_{\text{installation,offshore}-\text{substation}} \cdot C_{\text{day,HLV}} + C_{\text{mobilization,HLV}} + C_{\text{installation,onshore}-\text{substaion}}$$

The total installation costs for the electrical system ($C_{\text{installation,electrical}}$) is simply a summation of the installation costs for the cables and the substations.

$$C_{\text{installation,electrical}} = C_{\text{installation,cables}} + C_{\text{installation,substation}}$$

**O&M**

The total Operations and Maintenance (O&M) costs consist of the fixed operational costs ($C_{\text{operational}}$), the costs for preventive maintenance and corrective maintenance (for both turbines and balance of plant), and the salaries for technicians. Both corrective and preventive maintenance includes vessel costs and spare part costs.

The costs for preventive maintenance include the costs to inspect the turbine, the support structure, and the substations. This is done via Crew Transfer Vehicles (CTVs).

$$C_{\text{preventive}} = T_{\text{service}} \cdot N_{\text{CTV}} \cdot C_{\text{day,CTV}}$$

Corrective maintenance can be due to minor repairs, major repairs, or major replacements. Majority of the costs are due to major replacement since that requires the use of a WTIV. The number of vessel trips is equal to the number of instances of failure ($N_{\text{instances}}$), which, in turn, depends on the failure rate ($F$) and the number of turbines ($N_{\text{T}}$). For instance, the total time taken for major replacement ($T_{\text{replacement,major}}$) depends on the time taken by the vessel to travel to the site ($T_{\text{travel,WTIV}}$) and make the replacement ($T_{\text{repair,major}}$), and the total number of such failure instances in a year.

$$T_{\text{replacement,major}} = N_{\text{instances}} \cdot (T_{\text{repair,major}} + T_{\text{travel,WTIV}})$$

The total costs for major replacement for RNA, for instance, constitute the vessel costs and the spare part costs, expressed as a fraction of the RNA costs ($C_{\mathrm{RNA}}$), as shown below.

$$805 \quad C_{\mathrm{replacement,major}} = T_{\mathrm{replacement,major}} \cdot C_{\mathrm{day,WTIV}} + 0.1 \cdot C_{\mathrm{RNA}} \cdot N_{\mathrm{instances}}$$

Similarly, the costs of minor repairs, major repairs, and cable replacements can be determined using the respective failure rates, repair times, and spare part costs. Lastly, the cost of the technicians ($C_{\mathrm{technicians}}$) depends on the number of technicians, that are scaled from the reference with the number of turbines in the farm, and a fixed annual salary. The total O&M costs can then be evaluated as shown below.

$$810 \quad C_{\mathrm{OPEX}} = C_{\mathrm{operations}} + C_{\mathrm{preventive}} + C_{\mathrm{repair,minor}} + C_{\mathrm{repair,major}} + C_{\mathrm{replacement,major}} + C_{\mathrm{technicians}}$$

*Author contributions.* The research was carried out by Mihir under the guidance and supervision of Michiel and Dominic. The paper was collectively written by all three, and Michiel and Dominic had several valuable inputs and insightful discussions with Mihir about the work presented in this paper.

*Competing interests.* The authors declare that they do not have conflicts of interest.

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
