# Peer review of "Drivers for optimum sizing of wind turbines for offshore wind farms"

_Wind Energy Science, 2023_

## Referee Comment (RC1)

**Manuscript ID: WES-2023-39  Turbine scaling for offshore wind farms**

Authors: Mihir Kishore Mehta, Michiel Bastiaan Zaaijer, and Dominic von Terzi

In this work, using a wind farm cost model, the authors are demonstrating the possibility to find an optimum turbine scale for a given wind farm case. Cost and AEP gradients over rotor diameter and rated power are used to visualize the mutual effects within the wind farm. A sensitivity analysis on the model parameters (costs, farm characteristics) are presented. Lastly, different scenarios with respect to wind farm design constraints are investigated, which could be useful for wind farm developers.

The topic is of definite interest to the scientific community and is well written. I do have some comments and request of changes before the final publication.

**General comments:**

The title looks a bit too general; it could have been more specific.

- In general, the level of detail in the cost equations are low. But maybe this is fine. Authors have made the code public so the results should be reproducible, but maybe a short Appendix to the paper can help the reader to better understand some details.

*Section 2.2:*
- Line 101: T"he design space w.r.t. the two design variables, P and D, is **shown** in Fig. 2a, where the entire framework is run for the discrete set of points **shown**". Please review this sentence.
- Equation (2). The decommission costs $C_{DECOM}$ introduced in this equation are not explained in the text of this section (even not mentioned). Please provide a short description about these costs as done for the other.
- More in general: a NOMENCALTURE table can be useful
- Figure 2.a & 2.b. I understand that in extreme cases like 10MW turbine with 300m diameter rotor and 20MW turbine with 180m, rated wind speeds are set very low and very high, respectively. Maybe this point needs to be further elaborated in the text, making a connection to Weibull distribution and dependency on average wind speed, as we see later in the paper. It could be that for high/low wind sites, the design space should be adapted accordingly.
- More in general , is not clear where the data of Fig2b come from. Which data and model have been used to extract these LCoEs?

*Section 2.3.:*
- Line 113. Is not clear why the model uses non-dimensional thrust LUT and dimensional power LUT ($C_T$ and $P_{turbine}$).
- Line 116 & Eq. (3): "The ratio of the rated wind speeds ($V_{rated}$) compensates for any additional increase/decrease in the thrust due to a change in the rated wind speed for turbines with a specific power different from that of the reference turbine." This assumes that a linearly scaled aerodynamic and structural shape of the blade would result in the same thrust coefficient as the reference turbine. One can work with this assumption, but it needs to be clearly stated in the text.
- It is not clear to me if the rated wind speeds are changing when the rated power of the turbines are changed as they are scaled from reference. Even for two wind turbines with same specific powers, v_rated can be different if the resulting power coefficients are different. Finally, are

there some constraints on the maximum rotor speed, for instance to include some simplified noise-constraints) ?

**Section 2.3.4 Support Structure**

- Not very critical but it is not written which type of foundation is used in the model. The reader can presume that it is monopile, but I think it is better to make it clear.
- Line 157. The sizing is based on ultimate limit. Which are these limits? How the loads have been considered?

**Section 2.5**

- Cut-in cut-out wind speeds are not mentioned. It is not very critical, and they are probably kept constant, but it would be nice to add for completeness.

**Section 3.1**

- One could draw a line equivalent to the specific power 350 W/m2 investigated later in the paper, connecting the points calculated for that comparison. With that maybe it would also justify the selection of this specific power in section 3.1, showing that it coincides with the turbines investigated there. A separate figure LCoE vs specific power could also be added (like a scatter plot).

**Section 3.2.**

- A separate figure LCoE vs specific power could also be added (like a scatter plot). The points equivalent to the specific power 350 W/m2 investigated in Section 3.1 could be highlighted in this plot as well.
- Figure 8a: I think it is showing negative gradients, i.e. cost decrease per change in D and P.
- Equations 16, 17, 18. I honestly do not know this type of notation used to describe the derivatives. For example, the last one, $\frac{\partial AEP}{\partial D \partial P}$ looks wrong. From the right hand side of the equation it's clear that you are computing the FIRST derivative of AEP wrt ONE variable, so it should be $\frac{\partial AEP}{\partial D}$ and $\frac{\partial AEP}{\partial P}$, two different first order derivatives wrt to the two variables. Otherwise one should write $\frac{\partial^2 AEP}{\partial D \partial P}$ referring to the second derivative of the AEP wrt the two variables, i.e. $\frac{\partial}{\partial D}\left(\frac{\partial AEP}{\partial P}\right) = \frac{\partial}{\partial P}\left(\frac{\partial AEP}{\partial D}\right)$. But it's clear from the results of this derivate that the authors refer to the first case, i.e. the gradient wrt every single variable.

**Chapter 4:**

- Section 4.2 and 4.3 are a bit difficult to follow in general. A table where all these constraints and their impacts on major cost elements, AEP, wake losses etc. are summarized (with arrows going up/down etc.) could be nice.
- Figure 13: It seems graphically in Figure13b that the vector sum of gradients AEP' and Cost' from 'fixed-are-only' case would point to higher rated power and lower diameter. However, the new global optimum is located at higher rated power but also higher diameter. This shows that going in the direction of the gradient at baseline optimum would yield a local optimum. The explanations given in this section have different 'weightings' at every point in this design space. Maybe this point is worth emphasizing.

**Chapter 5 Conclusion**

- I am wondering if the presented results are sensitive (if yes, how sensitive) to the starting baseline design. We see that global optimum is found at 15MW rated power, which is equal to the rated power of the starting design IEA 15MW turbine with a smaller diameter. Can it be

that Cp Ct curves, which are calculated using baseline design, are causing a bias towards 15MW?

- **Line 586:** "However, while the optimum specific power is fairly sensitive to particular project conditions, this shift is less sensitive to scale (first bullet point)." **Second part of the sentence is not clear to me, rephrasing could be necessary.**

---

## Author Comment (AC1)

**Manuscript ID: WES-2023-39**
**Turbine scaling for offshore wind farms**

**September 24, 2023**

The authors would like to thank the reviewers for their time and valuable comments. The comments of both reviewers are useful as they improve the quality of the paper. The comments of each reviewer are addressed separately. The explanation is marked in blue while the actual changes in the manuscript are marked in red.

We would also like to mention that, since the paper submission, we have updated the modeling of the Rotor Nacelle Assembly (RNA) module to improve the consistency with the IEA 15MW reference turbine. As a consequence, the absolute numbers have changed, but all the trends are exactly the same, and all the conclusions still hold true. The updated values of LCoE for the entire design space, along with the new global optimum, are shown in Figure 1.

**Reviewer 1**

*The title looks a bit too general, it could have been more specific.*
We agree that the title can be a bit more specific.
The new title is now 'Drivers for optimum sizing of wind turbines for offshore wind farms'.

*In general, the level of detail in the cost equations are low. But maybe this is fine. Authors have made the code public so the results should be reproducible, but maybe a short Appendix to the paper can help the reader to better understand some details.*
The authors tried to explain the essential parameters that influence the costs for each discipline while the details can be found in the publicly available repository. However, as other readers may feel the same as reviewer 1, a short appendix has been added to provide explanations of some key models and provide additional equations used. This is also attached at the end of this document.

*A NOMENCLATURE table can be useful.*
A nomenclature was considered but, since there are not a lot of special symbols, it was deemed to not help a lot. We've tried our best to keep the symbols easily understandable and repeat the description in the text wherever thought to be helpful.

**Section 2.2**

*Line 101: "The design space w.r.t. the two design variables, P and D, is shown in Fig. 2a, where the entire framework is run for the discrete set of points shown." Please review this sentence.*
To make it more clear, it is now rephrased. 'The design space w.r.t. the two design variables, rated power (P) and rotor diameter (D) of the turbine, is shown in Fig. 2a. The entire wind farm level framework will be run for these discrete set of points'.

*Equation (2). The decommission costs $C_{\mathrm{DECOM}}$ introduced in this equation are not explained in the text of this section (even not mentioned). Please provide a short description about these costs as done for the other.*
Sentence rephrased to 'The numerator contains the Capital Expenditures ($C_{\mathrm{CAPEX}}$) that are paid

off initially, the summation of all the annual actualized Operation and Maintenance Expenditures ($C_{\text{OPEX}}$), and the costs to decommission the entire wind farm at the end of its lifetime ($C_{\text{DECOM}}$).'

*Figure 2.a and 2.b. I understand that in extreme cases like 10MW turbine with 300m diameter rotor and 20MW turbine with 180m, rated wind speeds are set very low and very high, respectively. Maybe this point needs to be further elaborated in the text, making a connection to Weibull distribution and dependency on average wind speed, as we see later in the paper. It could be that for high/low wind sites, the design space should be adapted accordingly.*

As rightly pointed out, these extreme designs will have either very high or low rated wind speeds. Since this section was more focused on simply introducing the design space, the specific comment about the rated wind speed was not mentioned. Also, for different sites, the design space adjusting might be more useful if the optimum is found to be at the boundary (which was not the case).

*More in general, is not clear where the data of Figure 2.b come from. Which data and model have been used to extract these LCoEs*

We see that this figure might actually be confusing to introduce in this section. The LCoE comes from the same set of models described later. The figure is actually just used for illustration purposes to show that the LCoE will first be evaluated for the discrete set of points followed by a surface fit to get a finer resolution. However, we understand that it may be confusing to show LCoE as an example already, even before explaining any models.

Hence, as a correction, the figure has now been removed, and the explanation about surface fit is only in the text.

**Section 2.3**

*Line 113. Is not clear why the model uses non-dimensional thrust LUT and dimensional power LUT ($c_{\text{T}}$ and $P_{\text{turbine}}$).*

*Line 116 and Eq. (3): "The ratio of the rated wind speeds ($V_{\text{rated}}$) compensates for any additional increase/decrease in the thrust due to a change in the rated wind speed for turbines with a specific power different from that of the reference turbine." This assumes that a linearly scaled aerodynamic and structural shape of the blade would result in the same thrust coefficient as the reference turbine. One can work with this assumption, but it needs to be clearly stated in the text.*

*It is not clear to me if the rated wind speeds are changing when the rated power of the turbines are changed as they are scaled from reference. Even for two wind turbines with same specific powers, $V_{\text{rated}}$ can be different if the resulting power coefficients are different. Finally, are there some constraints on the maximum rotor speed, for instance, to include some simplified noise-constraints)?*

These are great observations and we agree that the assumption about the aerodynamic properties being the same across all designs should be clearly mentioned. The assumptions about cut-in, cut-out and drivetrain efficiency are also mentioned. The entire paragraph is now rephrased.

The rotor aerodynamic performance is evaluated using the classic Blade Element Momentum (BEM) theory. The properties of a reference turbine are used as an input to determine the aerodynamic and structural performance and other RNA properties. The values of power coefficient ($c_{\text{P}}$) and thrust coefficient ($c_{\text{T}}$) in the partial load region are evaluated using the airfoil distribution, the normalized chord and twist profiles, and the tip speed ratio from the reference turbine. It should be noted that since the aerodynamic properties are the same as that of the reference, the resulting peak power and thrust coefficient are the same for all the designs. The cut-in wind speed, cut-out wind speed, and drivetrain efficiency are also held constant across all the designs. The rated wind speed of the turbine can then be determined followed by the evaluation of the power curve.

The paragraph before Equation (3) is also rephrased.

The rotor mass scaling model uses classical geometric scaling rules with an additional factor, as shown in Eq.(3), where $V_{\text{rated}}$ is the rated wind speed. Both the chord of the blade and the internal layup are scaled linearly with radius. Since the thrust coefficient is the same for all the designs, the additional increase/decrease in thrust is due to a change in the rated wind speed for turbines with a specific power different from that of the reference turbine. The ratio of the rated wind speeds compensates for any additional increase/decrease in the thrust, resulting in the same normalized (with rotor radius) tip deflection as that of the reference.

**Section 2.3.4**

*Not very critical but it is not written which type of foundation is used in the model. The reader can presume that it is monopile, but I think it is better to make it clear.*
Rephrased to, 'The sizing module used for the design of monopiles is based on the work of Zaaijer (2013).'

*Line 157. The sizing is based on ultimate limit. Which are these limits? How the loads have been considered?*
Rephrased to, 'The length of the transition piece is fixed while the monopile length is the sum of the evaluated monopile penetration depth, water depth, and the overlap with the transition piece. The aerodynamic and hydrodynamic loads and moments are calculated using the site characteristics and turbine data. The wind and wave loading is calculated for normal operation and also for extreme conditions with a 1-year and 50-year occurrence period. Additional safety factors are introduced for ultimate loads and to compensate for fatigue and buckling. The geometry for the tower and foundation is then determined by a root-finding algorithm that equates the calculated stresses to the permissible values.'

**Section 2.5**

*Cut-in cut-out wind speeds are not mentioned. It is not very critical, and they are probably kept constant, but it would be nice to add for completeness*
Now mentioned in the text. 'The cut-in wind and cut-out wind speed values of the reference turbine (3 ms$^{-1}$ and 25 ms$^{-1}$, respectively) are used for all turbine designs.'

**Section 3.1**

*One could draw a line equivalent to the specific power 350 W/m2 investigated later in the paper, connecting the points calculated for that comparison. With that maybe it would also justify the selection of this specific power in section 3.1, showing that it coincides with the turbines investigated there. A separate figure LCoE vs specific power could also be added (like a scatter plot).*
The aim of this section is just to show how various farm-level indicators change due to the change in turbine scale even for the same specific power. The choice of the specific power value per se is not so important there.
However, as suggested, it is interesting to plot the LCoE vs specific power, and this plot is now added to Section 3.2. The revised plot with the new baseline optimum (16.1MW-236m) is also shown below.

[Figure]

Figure 1: (a) Response map of LCoE w.r.t. the two design variables. (b) Optimum rated power and rotor diameter per constrained diameter and rated power, respectively. (c) LCoE plotted against the specific power

*Figure 8a: I think it is showing negative gradients, i.e. cost decrease per change in D and P*
As rightly pointed out, for the costs, the gradients are indeed negative and it is now mentioned in the text.
Gradients of costs and AEP components that include the weights A and B are indicated with an accent. The gradients at the LCoE optimum are shown in Fig. 8, where the cost gradients are negative and point in the direction of decreasing costs.

*Equations 16, 17, 18. I honestly do not know this type of notation used to describe the derivatives. For example, the last one, $\frac{\partial AEP}{\partial D \partial P}$ looks wrong. From the right hand side of the equation it's clear that you are computing the FIRST derivative of AEP w.r.t. ONE variable, so it should be $\frac{\partial AEP}{\partial D}$ and $\frac{\partial AEP}{\partial P}$, two different first order derivatives w.r.t. to the two variables. Otherwise one should write $\frac{\partial^2 AEP}{\partial D \partial P}$ referring to the second derivative of the AEP w.r.t. the two variables, i.e. $\frac{\partial}{\partial D}(\frac{\partial AEP}{\partial P}) = \frac{\partial}{\partial P}(\frac{\partial AEP}{\partial D})$. But it's clear from the results of this derivate that the authors refer to the first case, i.e. the gradient w.r.t every single variable*
As rightly pointed out, this was an error in the text, and apologies for the same. The authors indeed refer to the first case and it is now corrected. For the other equations, the derivate only w.r.t. D is shown and the text mentions that the gradients w.r.t. P can be similarly obtained.

$$\frac{\partial LCoE}{\partial D} = \frac{1}{AEP^2}\left(AEP \cdot \frac{\partial C}{\partial D} - C \cdot \frac{\partial AEP}{\partial D}\right) = A \cdot \frac{\partial C}{\partial D} - B \cdot \frac{\partial AEP}{\partial D}$$

$$\frac{\partial LCoE}{\partial P} = \frac{1}{AEP^2}\left(AEP \cdot \frac{\partial C}{\partial P} - C \cdot \frac{\partial AEP}{\partial P}\right) = A \cdot \frac{\partial C}{\partial P} - B \cdot \frac{\partial AEP}{\partial P}$$

**Chapter 4**

*Section 4.2 and 4.3 are a bit difficult to follow in general. A table where all these constraints and their impacts on major cost elements, AEP, wake losses etc. are summarized (with arrows going up/down etc.) could be nice.*
That's a good suggestion. The authors did consider making this revision. However, on further exploration, we found out that it may either not be very useful or a bit misleading when generalizing the results in a table. For instance, w.r.t. Section 4.2, an increase in farm power increases all costs and also the AEP, resulting in a lower LCoE, or an increase in the distance to the grid increases $C_{\text{CAPEX}}$ and hence, the LCoE. This may not be too useful in addition to the explanations in the text that also discuss how the changes drive the optimum. In the case of Section 4.3, generalizing the results could be a bit misleading. For instance, for the baseline problem (both farm power and area constraints), for a fixed number of turbines in the farm, the absolute spacing between the turbines is fixed while the normalized spacing is determined by the rotor diameter. However, for the fixed-power-only scenario, an assumption w.r.t. the normalized spacing is required (5D, in this study). Hence, the wake loss comparison between the two different constraint formulations cannot be generalized, even for the same turbine design, since wake losses depend on the turbine design itself and the assumption w.r.t. normalized spacing (for the fixed-power-only scenario).

*Figure 13: It seems graphically in Figure13b that the vector sum of gradients AEP' and Cost' from 'fixed-area-only' case would point to higher rated power and lower diameter. However, the new global optimum is located at higher rated power but also higher diameter. This shows that going in the direction of the gradient at the baseline optimum would yield a local optimum. The explanations given in this section have different 'weightings' at every point in this design space. Maybe this point is worth emphasizing.*
That's a fair point about the gradient pointing towards high power low diameter for the 'fixed-area only' case. It's just the direction of the gradient at the 'baseline optimum' and moving along that direction will lead to another point where the gradient direction will differ, ultimately leading to the

global optimum. It is simply the direction of the steepest descent/ascent at the baseline. The response surface is smooth and there are no local optimums. It is now clarified also in the text.

Fig. 13b shows the direction of the steepest descent/ascent at the baseline optimum, and moving along that direction will lead to another point where the gradient direction will differ, ultimately leading to the global optimum.

**Chapter 5: Conclusions**

*I am wondering if the presented results are sensitive (if yes, how sensitive) to the starting baseline design. We see that the global optimum is found at 15MW rated power, which is equal to the rated power of the starting design IEA 15MW turbine with a smaller diameter. Can it be that $c_P$ and $c_T$ curves, which are calculated using baseline design, are causing a bias towards 15MW?*

It's a great point and something that we have also considered for future work. The global optimum would definitely show some sensitivity to the chosen reference design since that forms the baseline for scaling of aerodynamic and structural properties (and costs) of the other designs. This is a reason why we evaluated the sensitivity to the rotor reference costs as well, in the model sensitivity aspect. The sensitivity, however, is not expected to be large. We would have been more worried about the chosen reference turbine in case the global optimum was further removed from the reference turbine. That could have indicated that the scaling laws would have been applied at or over their reasonable range of applicability. The earlier optimum rated power also being close to 15 MW (after the curve fit while the discrete optimum being at 14.5MW) is only for the chosen initial value of constraints and problem formulation. Any change in these values shifts the optimum. For instance, if the baseline farm power would have been 800 MW instead of the current value of 1 GW, the global optimum would have been around 14MW (as seen in the sensitivity to 'design inputs' section). However, after some modeling changes, the current baseline optimum is at 16.1MW-235m which again shows that the earlier optimum being at 15MW was a coincidence and not a bias. A change in the $c_P$ and $c_T$ values of the reference will have some impact on the optimum but don't seem to be the reasons to cause a bias towards the optimum rated power. However, conducting the sensitivity w.r.t. different reference designs (IEA 10 MW, the planned IEA 22 MW) will require a separate study and is on the list for future work, but out of scope for this manuscript. The point about the sensitivity to the reference design is now mentioned explicitly in the conclusions.

The findings in this research are obtained using low-fidelity cost models and the IEA 15 MW turbine as the reference design. However, the absolute values of the optimum will likely differ for a different reference turbine as the starting point for scaling, and a future study exploring the sensitivity to different reference designs with higher fidelity cost models is recommended. Nevertheless, the confidence in the use of scaling laws is largest when the scale of the reference turbine and of the global optimum are similar, as is the case here.

*Line 586: "However, while the optimum specific power is fairly sensitive to particular project conditions, this shift is less sensitive to scale (first bullet point)." The second part of the sentence is not clear to me, rephrasing could be necessary.*

The second part mentions how the project conditions (like wind speed, farm power, etc.) mainly drive the optimum specific power for any given scale of the turbine (like a 10 MW turbine or a 20 MW turbine). However, for better clarity, the sentence is now rephrased.

However, while the optimum specific power is fairly sensitive to particular project conditions, turbines of a fairly wide range of scales can perform equally well (first bullet point).

**Reviewer 2**

*The title of the paper does not convey the core contribution of this paper. Please be specific to include what authors' contributions to the research community are, besides what field authors are working in.*
We agree that the title can be a bit more specific.
The new title is now 'Drivers for optimum sizing of wind turbines for offshore wind farms'.

*The scientific implications of this study include the use of MDAO framework in the system-wide perspective of the wind farm design problem. Findings suggest that optimum specific power has different sensitivities to conditions and scale of the farm. Some findings may inform policy-makers and wind farm developers useful insights. However, the study itself is highly simplified and the design variables are highly limited. Thus, the reviewer think that this study is an exploratory work that provides the potential possibilities of application of MDAO in the large-scale wind farm design problems. The authors are suggested to clearly express this limitation in the context of paper scope.*
The authors agree that there are some simplifications in the modeling. The purpose of the research was to look at turbine sizing and the design variables (rated power and rotor diameter of the turbine) are limited but the most essential parameters that influence every aspect of the wind farm. Some models, although simplified, include the effects of changing the rated power or rotor diameter of the wind turbine on the LCoE of the wind farm. However, the limitation is now explicitly stated at the end of the introduction, and for more clarity, an appendix with some more equations has been added.
The turbine size refers to the two main defining variables of the turbine, rated power and the rotor diameter. The two variables are optimized w.r.t. the LCoE of a hypothetical wind farm, using an MDAO framework that includes low-fidelity models for every discipline of an offshore wind farm. The findings of this work may inform policy-makers and wind farm developers with useful insights. However, the implementation is simplified and the chosen set of design variables is limited. Thus, this study aims to be exploratory work that provides the potential possibilities of application of MDAO in large-scale wind farm design problems.

*In general, the reviewer views this manuscript missing many important information about modeling side. Please provide modeling details. All models used in this work are expressed in some function f, without any detail. Important models need to be articulated in the main content of the manuscript. If model is too large to be included, Appendix may be used. Without model details, the paper has little meanings to the research community.*
The aim of the modeling section was to mention the core dependencies on the input parameters, for each discipline. Since the code is open source (repository link mentioned in line 611), the results should be easily reproducible. However, we agree that some extra details can be provided in a short Appendix. The text of the added appendix is copied at the end of this document. Also, related to a comment about the AEP calculation, a line about the wake models is now added in the text.
'The Bastankhah Gaussian model along with the squared sum model is used to estimate wind speed deficits and the wake superposition, respectively.'

**XDSM**

*RNA is a nested sub-optimization problem, as given in Fig. 1. However, in the XDSM, the objective function for this optimization sub-problem is unclear. Also, which analysis model rerurns objective function value (or quantities used for calculating that) is unclear. It is obvious that the overall system-level design variables are P and D. However, for the RNA optimization sub-problem, what are the design variables, and how that connect to the system-level problem? Farm-level analysis model is not shown clearly. What factors drive difference in the objective function, when farm area, number of turbines, and each turbine scale changes? Within the XDSM given in Fig. 1, information regarding this aspect is largely missing. The XDSM given in Fig. 1 does not provide enough detail about models, how models interact with each other, and what information is exchanged between models. Please refer to the original XDSM paper (Lambe and Martins, 2012) and follow the widely-accepted XDSM conventions.*
That's a valid argument. The authors used the XDSM as an illustration to show the various disciplines (as analysis blocks) considered in the study and the final coupling variables used to calculate the objective function. The RNA was given the same color as the optimization as the two main design variables correspond to the RNA (rotor and the generator). However, it may be misleading since there

is no nested sub-optimization happening inside the RNA block. Apologies for the confusion. Also, as pointed out, it may be more helpful to show the input variables and some coupling variables between the models.
The XDSM is now changed and the new version is shown below.

[Figure]

Figure 2: eXtended Design Structure Matrix (XDSM) of the MDAO framework

**Other sections**

*Regarding Fig. 2(b), is there an identifiable point with minimum LCoE on the response surface? Also, please explain meaningful observations from this plot. The reviewer thinks that important interpretation on the observations here is largely missing.*
We see that this figure might actually be confusing to introduce in this section. The figure is actually just used for illustration purposes to show that the LCoE will first be evaluated for the discrete set of points followed by a surface fit to get a finer resolution. Hence, there are no insights or observations presented here in this section. However, we understand that it may be confusing to show LCoE as an example already, even before explaining any models.
Hence, as a correction, the figure is now removed, and the explanation of surface fit is mentioned in the text. The same LCoE surface is plotted as a contour plot and discussed in the results section.

*In Eq. (6), $N_T$ is discrete variable. How this discrete variable is incorporated in the optimization problem? Did the authors used mixed integer programming method to incorporate the discrete variables, used relaxation method to solve discrete problem in the continuous variable framework, or completely enumerated all possible discrete value options?*
This was briefly discussed in the part before Section 2.3 where the authors mention that the LCoE is evaluated for all possible discrete options shown in Figure 2.a. The number of turbines is not a design variable per se and is a result of the farm power and the rating of the turbine.

*The case studies involve a hypothetical site location with assumed environmental conditions in a stochastic manner. Please provide how these hypothetical site conditions are assumed or derived. If there is a reference for the probabilistic data, please provide a reference. Please also provide how much these hypothetical conditions represent the actual North Sea conditions in the real world, as the authors claimed the condition to be the hypothetical site in the North Sea.*
The site conditions per se are not hypothetical but the farm configuration is. For the wind resource,

the study uses ERA5 data at the coordinates of existing wind farms in the North sea. The other parameters like distance to the grid, water depth, etc. are also representative of existing wind farms in the North Sea. The references for the same are now explicitly added in Section 2.5

A hypothetical site and wind farm in the North Sea are considered. The site parameters and the farm orientation define the case study. The wind rose for the hypothetical site, shown in Fig. 3a, uses ERA5 Reanalysis data for a location near the Borselle wind farm in the North Sea (Hersbach et al., 2018). Some other case-defining parameters, like the distance to grid, water depth, etc., representative of tendered wind farms in the North Sea (RVO, 2016), are listed in Table 3.

*Regarding Eq. (15), (17), (18), the reviewer cannot understand the meanings of these equations. First of all, the equations are wrong. $\frac{\partial C}{\partial D \partial P}$ is mathematically not correct way for the second order partial derivatives. Please revisit these equations, and provide correct equations. Also, the reviewer cannot understand the physical meaning of the second-order partial derivatives in the context of gradient. They can be vector quantities with first-order partial derivatives to represent gradient. Or, they can represent Hessian in the context of positive definitiveness of the response surface. Eq. (15), (17), (18) do not represent either of them, and the reviewer cannot understand their meanings.*

Apologies for this error. The authors indeed refer to the first-order partial derivatives to represent gradient. It is already corrected for LCoE. For the other equations, the derivate only w.r.t. D is shown and the text mentions that the gradients w.r.t. P can be similarly obtained.

$$\frac{\partial LCoE}{\partial D} = \frac{1}{AEP^2} \left( AEP \cdot \frac{\partial C}{\partial D} - C \cdot \frac{\partial AEP}{\partial D} \right) = A \cdot \frac{\partial C}{\partial D} - B \cdot \frac{\partial AEP}{\partial D}$$

$$\frac{\partial LCoE}{\partial P} = \frac{1}{AEP^2} \left( AEP \cdot \frac{\partial C}{\partial P} - C \cdot \frac{\partial AEP}{\partial P} \right) = A \cdot \frac{\partial C}{\partial P} - B \cdot \frac{\partial AEP}{\partial P}$$

*Please use the same mathematical style throughout the entire manuscript. Check styles of Eq. (15)-(18), in comparison to Eq. (7). E.g., AEP, LCoE should not be italicized as they are not mathematical symbols, but abbreviated word representing quantities. Follow the style used in Eq. (7) for the entire manuscript. Check subscript styles of Eq. (17)-(18). "turbine", "other", "support", "wake", ..., they also need to be non-italized as they are not symbols. Follow the same style used in earlier equations, e.g., Eq. (7)-(14).*

As suggested, all the style corrections are made in the text.

*Regarding Fig. 10(b), why markers represent values in certain design directions only? They can have values in the entire D and P plane. Also, why the directions are different for each quantitie of interest? Please provide details of the authors' reasoning for the decisions.*

The current text gives some details about how each parameter affects the direction of the optimum. However, for more clarity, the following sentences are added.

[revised manuscript text omitted]

$$C_{\text{replacement,major}} = T_{\text{replacement,major}} \cdot C_{\text{day,WTIV}} + 0.1 \cdot C_{\text{RNA}} \cdot N_{\text{instances}}$$

Similarly, the costs of minor repairs, major repairs, and cable replacements can be determined using the respective failure rates, repair times, and spare part costs. Lastly, the cost of the technicians ($C_{\text{technicians}}$) depends on the number of technicians, that are scaled from the reference with the number of turbines in the farm, and a fixed annual salary. The total O&M costs can then be evaluated as shown below.

$$C_{\text{OPEX}} = C_{\text{operations}} + C_{\text{preventive}} + C_{\text{repair,minor}} + C_{\text{repair,major}} + C_{\text{replacement,major}} + C_{\text{technicians}}$$

---

## Author Response (AR2)

**Manuscript ID: WES-2023-39**
**Drivers for optimum sizing of wind turbines for offshore wind farms**

November 3, 2023

The authors would like to thank the reviewers for examining the revised manuscript, for the acceptance of the manuscript by reviewer 1, and for the valuable additional feedback from reviewer 2. The reply to the comment made by reviewer 2 is marked in blue while the actual changes in the manuscript are marked in red.

**Reviewer 2 comment**

*The authors have made an effort to address the technical concerns raised during the initial review. The manuscript has seen improvements based on the comments provided earlier. However, I am still contemplating the broader significance of this contribution to the research community. The study, as presented, seems to be an application of existing methodologies with a somewhat simplified representation of wind farm design. The analyses of the results could benefit from a deeper exploration of their significance. Recognizing the challenges of prolonging the review process, it would be enriching to observe distinct scientific values beyond minor improvements in LCoE and its associated sensitivity. Given the manuscript's simplified modeling approach, the current fidelity of the LCoE estimation might fall short in fully underscoring the study's merit. It would be beneficial if the authors could delve deeper into the advantages of the concurrent MDAO approach, particularly highlighting any unique optimal design outcomes that may not be evident through traditional methods.*

The authors understand that the significance of the study may not have been fully clear to the reviewer and may have been interpreted differently. This paper does not intend to focus on LCoE improvements but rather provides a new way of looking at drivers of optimum turbine sizes, via gradients of key farm-level parameters. The models used capture the necessary trade-offs at a farm level and hence, the paper doesn't draw insights about the absolute values themselves but rather, the trends. There have been studies that have looked at the benefits of MDAO in offshore wind exploring the benefits of MDAO at a turbine level using aeroelastic simulations. However, a comprehensive study looking at turbine sizing in an offshore wind farm capturing the essential trade-offs is missing in the literature. The paper first establishes that there is a global optimum beyond which, upscaling might not be beneficial. The paper also shows how upscaling, in general, along a constant specific power, does not result in LCoE reductions. This contributes significantly to the ongoing debate about whether or not to upscale.

The findings of this study have also been used in providing feedback for the proposal of the North Seas Standard (https://www.nwea.nl/the-north-seas-standard-enable-growth-with-wind-turbine-standardization/). The standards propose a minimum tip clearance of 25 m and maximum tip height of 1000 ft (aviation limit), restricting the maximum rotor diameter to be around 280 m. The study shows how imposing this limit is not a threat to LCoE reductions that can be attributed to turbine sizing. The LCoE of all the designs in the 300-400 $Wm^{-2}$ range differ by less than 5% compared to the LCoE of the global optimum (see Fig. 1). This again implies that continuous upscaling in the direction of similar specific power will lead to LCoE improvements that may be less than improvements obtained by the benefits that standardization may have on manufacturing, installation and supply-chain optimization.

[Figure]

Figure 1: LCoE across the entire design space

How the global optimum shifts with any change in the technology (material) or farm design conditions, or policies, can simply be understood by looking at the impact on specific gradients. The paper discusses how certain parameters (like wind speed, fixed costs) affect the gradient in the direction of changing specific power while others (like farm power density) drive a change in the direction of constant specific power. The gradients are also useful in analyzing how a change in the policy (in the form of removal of farm power/area constraint) shifts the optimum.

In the literature, a comprehensive study discussing how several factors impact both the magnitude and direction of the global optimum, is missing. It is essential to capture the necessary trade-offs at a farm level, making the usage of MDAO, for a turbine sizing study, inevitable. Hence, the intent was not to focus on how an MDAO-driven approach performs better than a traditional sequential approach. However, we understand that the benefits of such an analysis need to be better explained and the modeling limitations might not fully underscore the merits of this work. Hence, we think it could be useful to explicitly clarify, in the manuscript, that the purpose of the paper is not to show how an MDAO-based approach is superior but rather, to use it as a means to perform a comprehensive system-level analysis. A general overview of the approach used in this study along with the purpose of MDAO in this research is discussed in a new section (2.1), now added at the beginning of the methodology chapter. The research question is also slightly tweaked since the 'How to size...' might also be construed as 'what method to use' putting an emphasis on the method, which was not the original intent. Following the suggestion, a new section (3.3) is added before the sensitivity section that explicitly focuses on the benefits of having these individual gradients and their usefulness in determining the drivers for the optimum turbine size. Finally, some benefits of the study are explicitly stated at the end of the conclusions.

The research question is now formulated as:

[revised manuscript text omitted]

The purpose of the study and the approach is also clearly stated in the abstract now:

Large-scale exploitation of offshore wind energy is deemed essential to provide its expected share to electricity needs of the future. To achieve the same, turbine and farm-level optimizations play a significant role. Over the past few years, the growth in the size of turbines has massively contributed to the reduction in costs. However, growing turbine sizes come with challenges in rotor design, turbine installation, supply chain, etc. It is, therefore, important to understand how to size wind turbines when minimizing the Levelized Cost of Electricity (LCoE) of an offshore wind farm. Hence, this study looks at how the rated power and rotor diameter of a turbine affect various turbine and farm-level metrics

and uses this information in order to identify the key design drivers and how their impact changes with setup. A Multi-disciplinary Design Optimization and Analysis (MDAO) framework is used to perform the analysis. The framework uses low-fidelity models that capture the core dependencies of the outputs on the design variables while also including the trade-offs between various disciplines of the offshore wind farm. The framework is used, not to estimate the LCoE or the optimum turbine size accurately, but to provide insights into various design drivers and trends. A baseline case, for a typical setup in the North Sea, is defined where LCoE is minimized for a given farm power and area constraint with the IEA 15 MW reference turbine as a starting point. It is found that the global optimum design, for this baseline case, is a turbine with a rated power of 16 MW and a rotor diameter of 236 m. This is already close to the state-of-the-art designs observed in the industry and close enough to the starting design to justify the applied scaling. A sensitivity study is also performed that identifies the design drivers and quantifies the impact of model uncertainties, technology/cost developments, varying farm design conditions, and different farm constraints on the optimum turbine design. To give an example, certain scenarios, like a change in the wind regime or the removal of farm power constraint, result in a significant shift in the scale of the optimum design and/or the specific power of the optimum design. Redesigning the turbine for these scenarios is found to result in an LCoE benefit of the order of 1-2% over the already optimized baseline. The work presented shows how a simplified approach can be applied to a complex turbine sizing problem, that can also be extended to metrics beyond LCoE. It also gives insights to designers, project developers, and policy makers as to how their decision may impact the optimum turbine scale.